# Rashba-like spin splitting along three momentum directions in trigonal layered PtBi$_2$

Ya Feng[1,2,10], Qi Jiang[3,10], Baojie Feng [4,5,10], Meng Yang [4,10], Tao Xu [3,6], Wenjing Liu[3], Xiufu Yang[1], Masashi Arita[2], Eike F. Schwier[2], Kenya Shimada [2], Harald O. Jeschke [7], Ronny Thomale[8], Youguo Shi[4], Xianxin Wu[8]*, Shaozhu Xiao[1]*, Shan Qiao[3,6] & Shaolong He [1,9]*

Spin-orbit coupling (SOC) has gained much attention for its rich physical phenomena and highly promising applications in spintronic devices. The Rashba-type SOC in systems with inversion symmetry breaking is particularly attractive for spintronics applications since it allows for flexible manipulation of spin current by external electric fields. Here, we report the discovery of a giant anisotropic Rashba-like spin splitting along three momentum directions (3D Rashba-like spin splitting) with a helical spin polarization around the $M$ points in the Brillouin zone of trigonal layered PtBi$_2$. Due to its inversion asymmetry and reduced symmetry at the $M$ point, Rashba-type as well as Dresselhaus-type SOC cooperatively yield a 3D spin splitting with $\alpha_R \approx 4.36$ eV Å in PtBi$_2$. The experimental realization of 3D Rashba-like spin splitting not only has fundamental interests but also paves the way to the future exploration of a new class of material with unprecedented functionalities for spintronics applications.

[1] Ningbo Institute of Materials Technology and Engineering, Chinese Academy of Sciences, 315201 Ningbo, China. [2] Hiroshima Synchrotron Radiation Center, Hiroshima University, Higashi-Hiroshima, Hiroshima 739-0046, Japan. [3] State Key Laboratory of Functional Materials for Informatics and CAS Center for Excellence in Superconducting Electronics (CENSE), Shanghai Institute of Microsystem and Information Technology, Chinese Academy of Sciences, 200050 Shanghai, China. [4] Institute of Physics, Chinese Academy of Sciences, 100190 Beijing, China. [5] School of Physical Sciences, University of Chinese Academy of Sciences, 100049 Beijing, China. [6] School of Physical Science and Technology, ShanghaiTech University, 200031 Shanghai, China. [7] Research Institute for Interdisciplinary Science, Okayama University, Okayama 700-8530, Japan. [8] Institute for Theoretical Physics and Astrophysics, Julius-Maximilians University of Wurzburg, Am Hubland, D-97074 Wurzburg, Germany. [9] Center of Materials Science and Optoelectronics Engineering, University of Chinese Academy of Sciences, 100049 Beijing, China. [10] These authors contributed equally: Ya Feng, Qi Jiang, Baojie Feng, Meng Yang. *email: xianxinwu@gmail.com; xiaoshaozhu@nimte.ac.cn; shaolonghe@nimte.ac.cn

The generation, manipulation, and detection of spin currents are the fundamental aspects of spintronics. The present material realizations for potential spintronics applications are based on spin–orbit coupling (SOC)[1,2], which is a relativistic interaction of the electron spin with its motion in a potential. It allows for the removal of spin degeneracy in a non-magnetic material without the use of an external magnetic field. Among different types of SOC, Rashba-type SOC[3,4] has attracted the most attention, since it can be manipulated by external electric fields, which is of central importance for applications in spintronics. In a system with Rashba SOC, each spin-degenerate parabolic band splits into two parabolic bands with opposite spin polarization. The band dispersions can be described by $E^{\pm}(\mathbf{k}) = (\hbar^2 \mathbf{k}^2/2m^*) \pm \alpha_R |\mathbf{k}|$, where $m^*$ is the effective mass of the electron and the Rashba parameter $\alpha_R$ represents the strength of the SOC. The Rashba-type SOC is usually caused by structure inversion asymmetry (SIA) which stems from the inversion asymmetry of the confining potential[2,3].

By contrast, there is other asymmetry in real materials called bulk inversion asymmetry (BIA). It originates from lack of inversion symmetry in the bulk and yields Dresselhaus-type SOC[5,6]. In many materials, these two kinds of SOC couple together, resulting in an anisotropy of spin splitting. Such anisotropy can lead to many interesting phenomena, such as a persistent spin helix observed in GaAs low-dimensional systems[7,8], long spin relaxation times[9–14], new device proposals, and more[15–19].

Rashba splitting was first observed by spin-resolved and angle-resolved photoemission spectroscopy (Spin-ARPES) in the Shockley surface state of Au(111)[20–22]. Many phenomena related to Rashba-type SOC have been studied in a variety of systems[23–50]. In particular, BiTeI and GeTe gained attention for their giant Rashba splitting in bulk states[35–45]. However, in all known instances, the Rashba bands remain spin degenerate along out-of-plane direction $k_z$ for $k_x = k_y = 0$, while they show spin splitting along the in-plane momentum directions, namely, $k_x$ and $k_y$ directions. While there is no principal symmetry exclusion argument against Rashba spin splitting along three momentum directions (3D Rashba spin splitting) induced by inversion symmetry breaking, its realization in an explicit material framework persists to be a challenging task.

In this work, we report the discovery of a giant anisotropic 3D Rashba-like spin splitting directly observed by spin-ARPES in a binary compound: PtBi$_2$. The spin-resolved ARPES employed the recently developed multichannel VLEED-type detector enabling fast mapping of a complete spin-polarized $E(\mathbf{k})$ map[51]. The magnitude of the spin splitting is even stronger than that of BiTeI[35]. The observed spin splitting originates from the breaking of inversion symmetry of the PtBi$_2$ crystal structure(space group $P31m$). In particular, the large spin splitting emerges at the $M$ points of the Brillouin zone instead of the $\Gamma$ point, where Rashba-type splitting is usually found. This bears crucial implications, rendering PtBi$_2$ unique among related material classes. The reduced point group symmetry at $M$ along with the BIA allow the Dresselhaus-type and Rashba-type SOC terms to cooperatively yield a large 3D anisotropic spin splitting with a helical texture around the $M$ point.

## Results

**The crystal structure and Fermi surface of PtBi$_2$.** For trigonal layered PtBi$_2$, two structural phases have been reported: space group $P\bar{3}$[52] and $P31m$[53,54]. Both phases are layered structures with each layer consisting of edge-sharing PtBi$_6$ octahedra, resembling the 1H-polytype of CdI$_2$. ARPES results on the $P\bar{3}$ phase have been reported very recently[55,56] motivated by the

discovery of giant linear magneto-resistance[57]. On the other hand, in the $P31m$ case, the distortion breaks inversion symmetry but the mirror symmetry is retained. Our X-ray single-crystal structure analysis demonstrates that the PtBi$_2$ grown and studied in this work belongs to the $P31m$ space group with lattice parameters $a = b = 6.5705$ Å and $c = 6.1707$ Å as shown in Fig. 1a, b (for XRD results, see supplementary). From the crystal structure in Fig. 1b, we can see that the Bi atoms in the layer above Pt atoms are coplanar while those below Pt are not. This leads to ABC stacking of three inequivalent layers, clearly breaking inversion symmetry. The corresponding bulk Brillouin zone is drawn in Fig. 1c in blue, and the projected surface Brillouin zone is sketched in orange. Figure 1d shows the experimental constant energy contour of PtBi$_2$ at $E_b = 440$ meV and around $k_z = 0$, which is in reasonable agreement with the calculation for $E_b = 420$ meV as shown in Fig. 1e. There are two six-pointed-star-shaped pockets around the $\Gamma$ point and one triangular pocket at the $K$ point. The elliptical contour around the $M$ point and large closed pocket encircling much of the Brillouin zone arise from anisotropic spin splitting. In the following, we will focus on the band structure around the $M$ point.

**Rashba-like band splitting at the $M$ point.** The band dispersion along the $\Gamma$–$M$–$\Gamma$ direction as measured by ARPES is displayed in Fig. 2a. It is evident that the bulk states exhibit large Rashba-type splitting with the crossing point at about $E_b = 520$ meV. This observed splitting is consistent with theoretical calculations as shown in Fig. 2b. The calculated band structure of PtBi$_2$ along other special $k$ lines can be found in Supplementary Fig. 2a. The peak positions of the momentum distribution curves (MDCs) corresponding to these split bands are compared against band calculations in Fig. 2b—we find good agreement between the experimental and theoretical band dispersions. From the experimental band dispersion, we can evaluate the characteristic parameters quantifying the strength of the Rashba splitting. Along the $M$–$\Gamma$ direction, the momentum offset $k_0 = 0.045 \pm 0.001$ Å$^{-1}$ and the Rashba energy $E_R = 98 \pm 2$ meV. Using $E_R = \hbar^2 k_0^2/2m^*$ and $k_0 = m^* \alpha_R/\hbar^2$, we have obtained $\alpha_R = 2E_R/k_0 = 4.36 \pm 0.14$ eV Å in PtBi$_2$. This value is even larger than that in BiTeI ($\alpha_R = 3.85$ eV Å)[35] and is one of the largest bulk Rashba states occurring at the lower symmetry $M$ points among previous publications.

Constant energy contours around the $M$ point for selected binding energies are displayed in Fig. 2c–k to demonstrate the dispersion of the split band. We show constant energy contours starting from $E_b = 400$ meV instead of the Fermi level to examine details near the crossing point at the $M$ point. Two contours appear at $E_b = 400$ meV: the inner contour is a closed ellipse but the outer one is open, in sharp contrast to the typical two concentric circles from pure Rashba splitting. With increasing binding energy, both the inner and outer contours tend to shrink gradually. The inner one shrinks to a point at $E_b = 520$ meV and the outer open one becomes two ellipses along $k_x$ ($\Gamma$–$M$ direction). At higher binding energies, the ellipses continue to shrink until they completely disappear for binding energy about 620 meV.

**Spin polarizaion of the Rashba-like splitting.** One characteristic feature of the Rashba splitting is a helical spin structure, which we confirm through spin-resolved ARPES measurements. The spin polarization $P_y$ along $\Gamma$–$M$–$\Gamma$ ($k_x$) obtained by substraction of spin-down ($s_y < 0$) and spin-up ($s_y > 0$) intensities is shown in Fig. 3a. Here $s_y$ denotes the spin component along the direction perpendicular to the slit of the electron analyzer (which is parallel to $\Gamma$–$M$–$\Gamma$ or $k_x$) and the surface normal. Red represents $P_y > 0$ and blue represents $P_y < 0$. Below $E_b = 400$ meV, it is

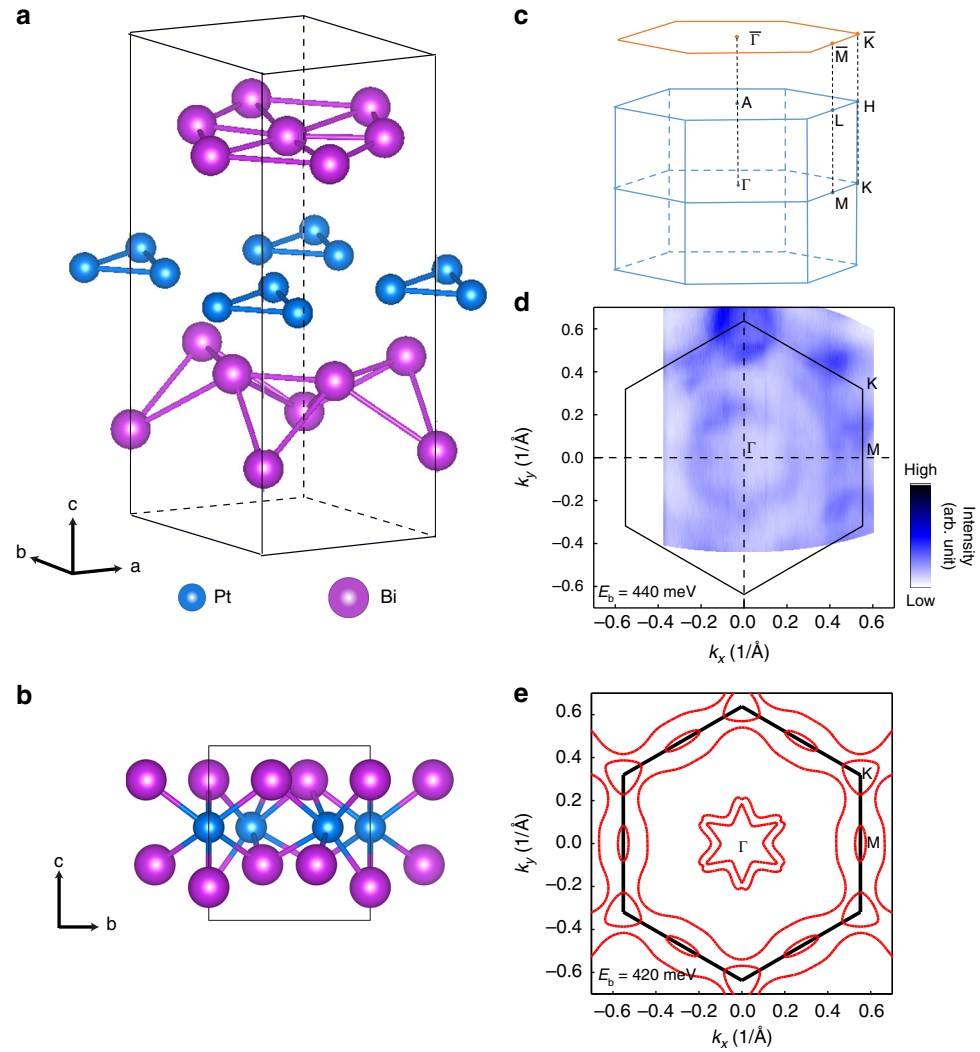

**Fig. 1** The crystal structure and Fermi surface of PtBi$_2$. **a**, **b** Crystal structure of trigonal layered PtBi$_2$ with space group $P31m$. **c** Bulk Brillouin zone (blue) and surface Brillouin zone (orange). **d** Constant-energy contour at $E_b = 440$ meV and around $k_z = 0$ measured by angle-resolved photoemission spectroscopy (ARPES) ($h\nu = 8.8$ eV). The color scale is linear as in all other images. **e** Calculated constant energy contour at $E_b = 420$ meV and $k_z = 0$

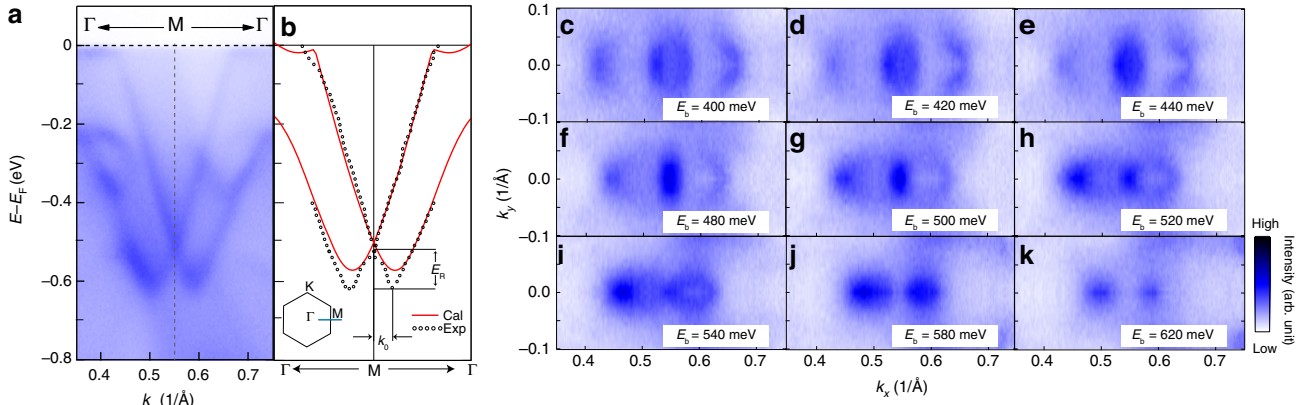

**Fig. 2** Rashba-like band splitting around the $M$ point. **a** Rashba-like band splitting in PtBi$_2$ at the $M$ point, shown along the $\Gamma$–$M$–$\Gamma$ direction (indicated by the blue line in the inset of **b**) measured by angle-resolved photoemission spectroscopy (ARPES) ($h\nu = 9$ eV). **b** Black circles show peak positions of the momentum distribution curves (MDCs) extracted from the ARPES data in **a**, while red lines represent the corresponding calculated bulk bands. The left inset shows the surface Brillouin zone (BZ) of PtBi$_2$. **c**–**k** Constant-energy contours obtained by integrating the photoemission spectral weight over a 10 meV energy window at binding energies as labeled

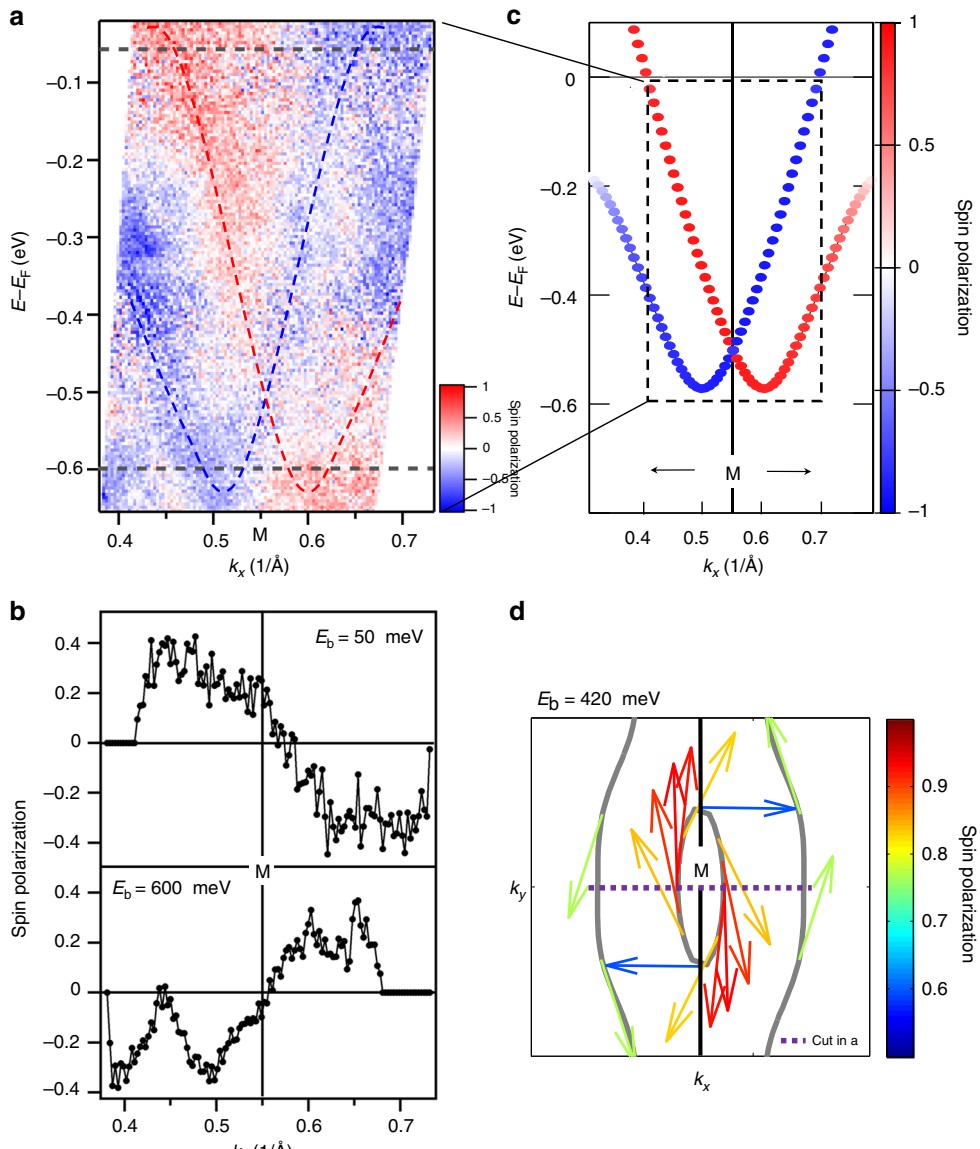

**Fig. 3** Spin texture of the Rashba-like splitting. **a** Spin-resolved band image around the $M$ point along $\Gamma$–$M$–$\Gamma$ ($k_x$) measured by spin-resolved and angle-resolved photoemission spectroscopy (SARPES) ($h\nu = 8.4$ eV). The red and blue lines indicate the locations of the Rashba-split bands, while the horizontal gray dashed lines indicate the energy positions of the momentum distribution curves (MDCs) in **b**. **b** MDCs of spin polarization at $E_b = 50$ and 600 meV extracted from the data in **a**. **c** Calculated spin polarization of the Rashba-split bands ($k_y = k_z = 0$). **d** Spin texture on the constant-energy contours around $M$ point at $E_b = 420$ meV. Arrows indicate the spin direction while their color indicates the degree of spin polarization. It has been checked that the experimental data in **b** exhibits the same spin chirality as the calculated data in **d**. The scale bars in **a**, **c**, and **d** represent spin polarization

apparent that the left(right) band dispersion is dominated by spin-down(up) intensity. These features are even more clearly demonstrated in Fig. 3b, which shows the momentum distribution curves (MDCs) of spin polarization at binding energies of 50 and 600 meV. The calculated spin-resolved band structure is displayed in Fig. 3c, which shows good agreement between the theoretical calculation and experimental data. Figure 3d shows the calculated spin texture on the constant-energy contour at $E_b = 420$ meV, where the arrows show the in-plane orientation of spin and their color indicates the degree of spin polarization. Both the inner ellipse and outer open contours around $M$ exhibit a helical spin structure. The spin polarization in experiments is by a factor of 2–3 lower than in the calculations. This was usually observed and might be related to the $k$-space resolution in experiments, influence of the inelastic background, and sample inhomogeneity.

**Origination of the Rashba-like splitting.** Comparing the band splitting along $M$–$\Gamma$ with $M$–$K$ in theoretical calculations, we find that the splitting has a significant anisotropy, which results in the observed elliptic contours. While an anisotropic effective mass could lead to anisotropic splitting, anisotropy in the underlying band structure is unable to explain why there are minima only along $k_x$. Only including pure Rashba SOC cannot explain the observed anisotropy at all. The band splitting in PtBi$_2$ occurs around the $M$ point, whose little group has lower symmetry than that of the $\Gamma$ point, where splitting is more commonly encountered. The mirror planes do not pass through $M$, and furthermore there are three inequivalent $M$ points. The lower symmetry allows the coexistence of Rashba-type and Dresselhaus-type SOC terms, while the additional degrees of freedom can increase the carrier concentration and introduce additional hybridization between the bands from different $M$ points. The little group is $C_s$ at the $M$

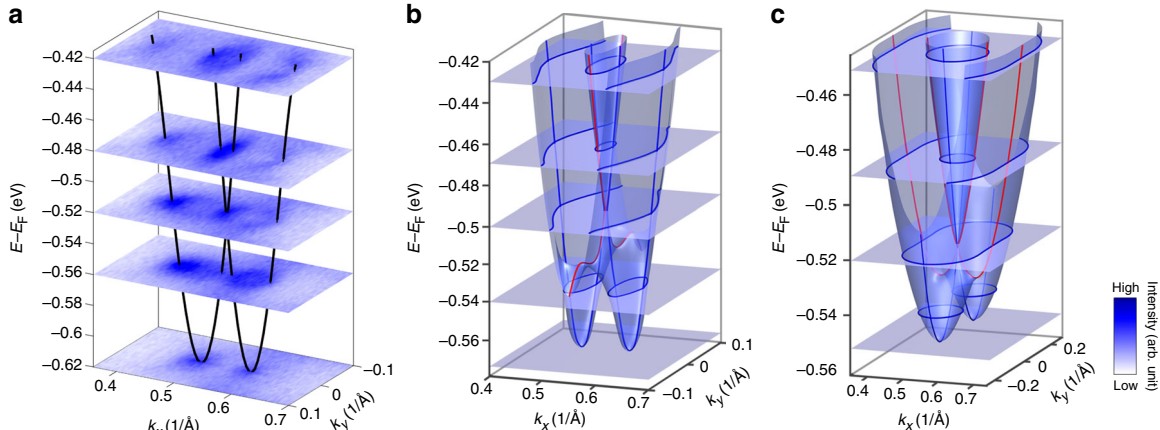

**Fig. 4** 3D electronic structure of the Rashba-like splitting. **a** 3D $E(k_x, k_y)$ map from angle-resolved photoemission spectroscopy (ARPES) data ($h\nu = 9$ eV). The black lines indicate the Rashba-split bands. Different surfaces correspond to constant-energy contours at different binding energies. **b** Calculated 3D electronic structure of PtBi$_2$ at the $M$ point. The red lines indicate the Rashba-split dispersion along $M$–$K$ ($k_y$) direction, while the blue lines indicate the Rashba-split dispersion along $M$–$\Gamma$ ($k_x$) direction. **c** Calculated 3D electronic structure considering only pure Rashba SOC, taking effective masses from the ARPES data. The blue lines indicate the Rashba-split dispersion along $M$–$K$ ($k_y$) direction, while the red lines indicate the Rashba-split dispersion along $M$–$\Gamma$ ($k_x$) direction. Both red (blue) lines in **b**, **c** indicate dispersions with smaller (larger) band splitting. The gray surfaces with blue lines on them in **b**, **c** sketch out the constant-energy contours at different binding energies

point and by using the theory of invariants the three-dimensional (3D) effective model around the $M$ point can be obtained. The effective model up to second order in **k** reads

$$h_{M_1}(\mathbf{k}) = \frac{k_x^2}{2m_x} + \frac{k_y^2}{2m_y} + \frac{k_z^2}{2m_z} + \alpha_1(\sigma_x k_y - \sigma_y k_x) + \beta_1(\sigma_x k_y + \sigma_y k_x)$$
$$+ \alpha_2(\sigma_x k_z - \sigma_z k_x) + \beta_2(\sigma_x k_z + \sigma_z k_x), \quad (1)$$

where the first three terms are the quadratic and the other terms correspond to the Rashba ($\alpha$) and Dresselhaus ($\beta$) SOC in the $k_x$–$k_y$ ($\alpha_1, \beta_1$) and $k_x$–$k_z$ planes ($\alpha_2, \beta_2$). The absence of rotational symmetry in the little group allows the coexistence of the Rashba and Dresselhaus terms. It is usually forbidden when the Rashba splitting occurs around $\Gamma$, taking BiTeI as an example[35].

For the $k_z = 0$ plane, the effective model is reduced to $h_{M_1}(k_x, k_y, 0) \frac{k_x^2}{2m_x} + \frac{k_y^2}{2m_y} + \alpha_1(\sigma_x k_y - \sigma_y k_x) + \beta_1(\sigma_x k_y + \sigma_y k_x) + (\beta_2 - \alpha_2)\sigma_z k_x$. The density functional theory (DFT) calculated Rashba parameter along $\Gamma M$ $\alpha_R$ is 3.15 eV Å, which is smaller than the experimental value. By fitting the DFT bands, we obtain $1/m_x = 45.0$ eV Å$^2$ ($m_x = 0.17m_e$), $1/m_y = 9.0$ eV Å$^2$ ($m_y = 0.85m_e$), $\alpha_1 = 1.33$ eV Å and $\beta_1 = -0.7$ eV Å. Therefore, Dresselhaus-type SOC cannot be neglected. In our trigonal lattice, each of the three $M$ points will contribute to elliptic Rashba-like bands. When the outer bands from different $M$ points meet, they can hybridize and form a large pocket around $\Gamma$ (open curves around $M$ point, see Supplementary Fig. 5). This explains the observed outer open contours around $M$ at binding energies of ~400–440 meV in Fig. 2c–e. In our calculations, we also find a small but non-negligible out-of-plane spin component, which is attributable to the SOC terms in the $k_x$–$k_z$ plane in the effective model.

We also show a 3D $E(k_x, k_y)$ map of the Rashba-like spin split bands as determined by ARPES in Fig. 4a and the 3D electronic structure calculated within DFT in Fig. 4b. Selecting constant-energy contours at certain binding energies for comparison, we find reasonable agreement. To further demonstrate the effect of Dresselhaus-type SOC, we calculated the band dispersions considering only the Rashba-type SOC term but taking the anisotropic effective masses determined by ARPES. The obtained bands are shown in Fig. 4c. The most pronounced feature is that the

band splitting along $k_x$ (red) is smaller than that along $k_y$ (blue). The two elliptic contours at large binding energy are located along the $M$–$K$ direction (not $M$–$\Gamma$ as in experiments). In our fit using both Rashba and Dresselhaus components, $\alpha_1$ and $\beta_1$ have the opposite sign. The Dresselhaus term will reduce the effective Rashba parameter ($\alpha_1 + \beta_1$) along $M$–$K$ but enhance it as $(-\alpha_1 + \beta_1)$ along $M$–$\Gamma$. This enhancement overcomes the effect of effective mass anisotropy, leading to a large band splitting along $M$–$\Gamma$. Therefore, we conclude that the splitting around the $M$ point in trigonal layered PtBi$_2$ is a result of the cooperation of Rashba-type and Dresselhaus-type SOC. The reasonable agreement between the measured and calculated bands indicates that the splitting originates from bulk inversion symmetry breaking, not surface effects.

**Three-dimensional nature of the Rashba-like splitting.** In order to investigate whether the Rasha-type spin splitting band is a bulk band or a surface state, we performed photon energy-dependent ARPES measurements with more than 15 different photon energies. In Fig. 5a–d, we selectively present bands taken along $\Gamma$–$M$ direction with photon energy of 9, 10, 11, 12 eV. These bands show strong photon energy dependence, indicating strong $k_z$ dispersion. Based on the photon energy-dependent ARPES measurements, we can obtain the band dispersion along $M$–$L$ direction in the three-dimensional Brillouin zone sketched in the inset of Fig. 5e. In order to obtain band dispersion along $M$–$L$ direction, the extracted energy distribution curves at the $M$ point taken at different photon energies are plotted as function of $k_z$ in Fig. 5e. In Fig. 5e, the band along $M$–$L$ direction also shows similar band splitting comparable to that along $\Gamma$–$M$ direction in Fig. 2. The observed maximum energy splitting is around 160 meV, which is smaller than the 384 meV along $\Gamma$–$M$ direction. It is worth noting that the value of 160 meV along $M$–$L$ direction is already lager than that of Rashba splitting observed in the surface state of Au(111)[20]. The obtained band splitting along $k_z$ direction can be qualitatively captured by the calculated results as shown in Fig. 5f. The specific shape of the dispersions is a little different, which may be attributed to the poor momentum resolution along out-of-plane direction in ARPES measurements. More importantly, our calculated results also verify that the split band is also spin polarized. Our results indicate that the band around the $M$ point not only shows spin splitting in the $k_x$–$k_y$ momentum plane

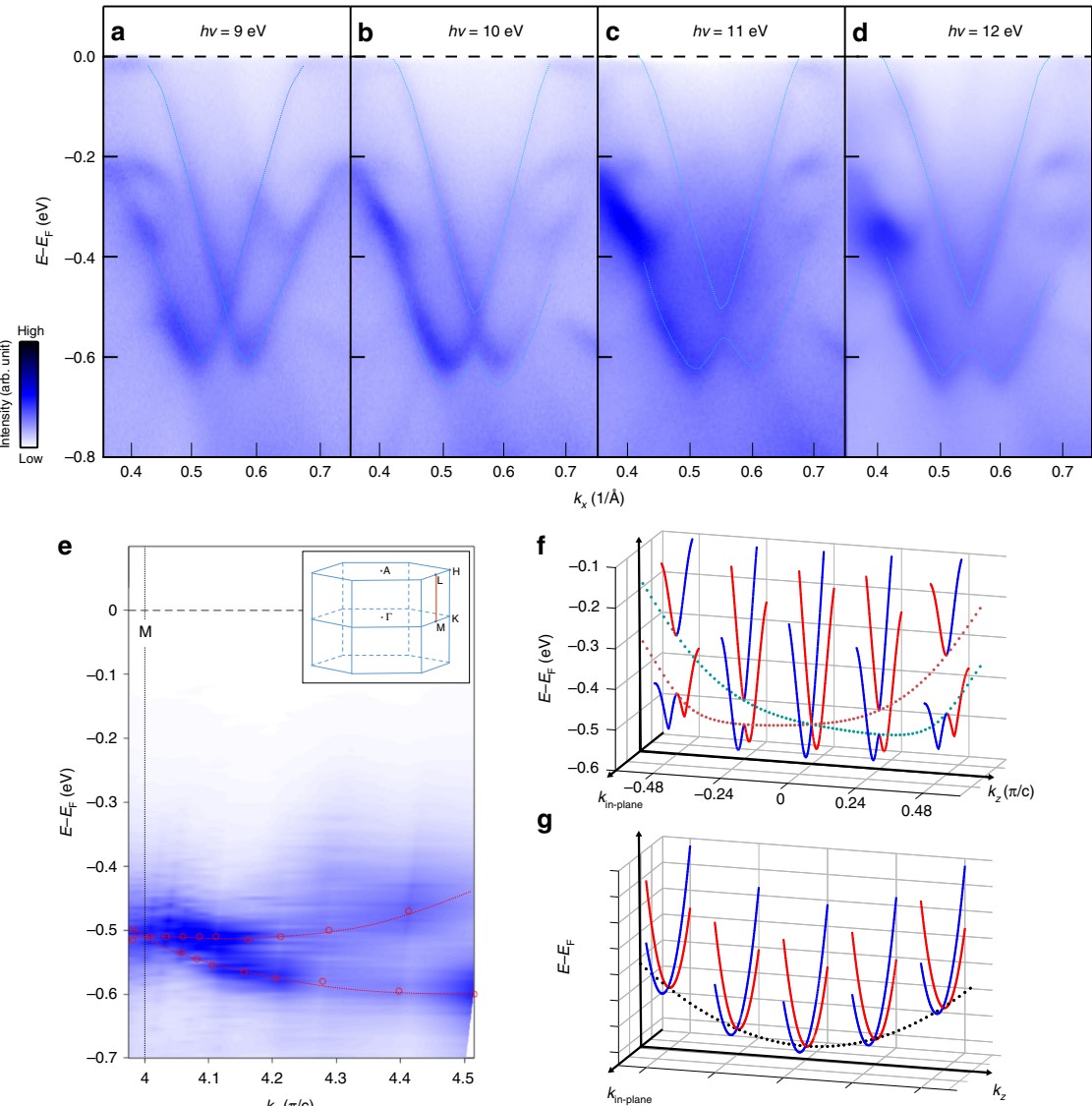

**Fig. 5** Three-dimensional nature of Rashba-like bands around $M$ point. **a–d** Energy–momentum image mapped by angle-resolved photoemission spectroscopy (ARPES) taken at different photon energies. **e** Band dispersion along $M$–$L$ direction obtained by plotting energy distribution curves at the $M$ point taken at different photon energies as function of $k_z$ (the $M$–$L$ direction as sketched in the three-dimensional Brillouin zone shown in the inset). **f** Calculated band structure of PtBi$_2$ along the in-plane and out-of-plane momentum directions. The red and blue solid lines show the in-plane band structures at different $k_z$. The red and green dashed lines show the band dispersions along $k_z$ direction, indicating Rashba splitting along $k_z$. **g** Sketch of Rashba bands which have dispersion along $k_z$ but remain spin degenerate along $k_z$ for $k_x = k_y = 0$

but also in the $k_x$–$k_z$ plane, which is also consistent with the effective model in Eq. (1). The smaller potential asymmetry with respect to the $xz$ plane, compared with the $xy$ plane, leads to a smaller strength of Rashba splitting along $k_z$. Therefore, the observed Rashba-like spin splitting in PtBi$_2$ is of three-dimensional nature: a consequence of coexistence of both Rashba-type and Dresselhaus-type SOC effects in $xy$ and $xz$ planes. So far, almost all of the reported Rashba bands only show spin splitting along the in-plane momentum directions. Although the Rashba bands of BiTeI and GeTe show dispersion along the out-of-plane direction ($k_z$)[36–38,40–42,45], they remain spin degenerate along $k_z$ for $k_x = k_y = 0$ as sketched in Fig. 5g, indicating a 2D Rashba spin splitting. Our results therefore represent the first realization of the three-dimensional Rashba-like spin splitting in a real material and potentially unfold numerous promising applications.

## Discussion

We have discovered a giant anisotropic 3D Rashba-like spin texture around the $M$ points in a binary compound: PtBi$_2$. The band splitting and its helical spin polarization measured by spin-ARPES are in good agreement with DFT calculations which also confirm the bulk nature of the effect. Due to the low point group symmetry $C_s$ at the $M$ point, the conspiring effects of Rashba-type and Dresselhaus-type SOC are essential to explain the observed 3D spin splitting. These results hence enrich our understanding of Rashba physics and inspire future exploration of new materials systems, which may host 3D Rashba spin texture and hold potential applications in spintronics.

## Methods

**Sample preparation**. Single crystals of noncentrosymmetric trigonal layered PtBi$_2$ were synthesized by the Bi self-flux method. Elemental Platinum (pieces, Alfa

Aesar, 99.99%) and bismuth (grains, PRMat, 99.9999%) were mixed in the molar ratio Pt:Bi = 1:6 and loaded into an alumina crucible in an argon-filled glovebox. The crucible was sealed in a quartz tube under high vacuum. The tube was heated to 1073 K and maintained for 20 h, then cooled slowly to 723 K at a rate of 3 K h$^{-1}$. The samples were then separated from the flux in a centrifuge and quenched immediately in cold water. An X-ray diffraction (XRD) pattern from the large surface of a PtBi$_2$ single crystal is shown in Supplementary Fig. 1a, indicating the crystal surface is parallel to the $ab$ plane. A low-energy electron diffraction (LEED) pattern of a cleaved PtBi$_2$ single crystal surface is shown in Supplementary Fig. 1b, confirming the trigonal symmetry and high quality of the PtBi$_2$ crystal. The structure is determined by performing single-crystal X-ray diffraction measurements(see supplementary).

**The ARPES measurements**. ARPES measurements were carried out on beamline BL-9A of the Hiroshima Synchrotron Radiation Center (HiSOR) using both synchrotron radiation (photon energy $hv = 8$–12 eV) and a Xenon lamp (photon energy $hv = 8.4$ eV) for photoelectron excitation sources. The total energy resolution was set to 20 meV and the base pressure was better than $5 \times 10^{-11}$ mbar. Spin-resolved ARPES measurements were carried out at the Shanghai Institute of Microsystem and Information Technology using a Xenon lamp and an imaging-type multichannel VLEED spin polarimeter[51]. The spin-resolved energy resolution was set to about 27 meV and the Sherman function $S_{eff}$ was ~0.25. For all ARPES measurements, the samples were cleaved in situ and measured at a temperature of about 40 K. The Fermi level was determined by the measurement of a poly-crystalline gold sample in electrical contact with the sample. All the experimental results were reproducible in multiple samples.

**Theoretical calculation**. Theoretically, we employed relativistic first-principle calculations based on the density-functional theory as implemented in the QUANTUM ESPRESSO code[58]. The core electrons were treated using the projector augmented wave method[59], and the exchange correlation energy was described by the generalized gradient approximation (GGA) using the PBE functional[60]. The cutoff energy for expanding the wave functions into a plane-wave basis was set to 50 Ry. The Brillouin zone was sampled in **k** space within the Monkhorst–Pack scheme[61] with a **k** mesh of $8 \times 8 \times 8$, which achieved reasonable convergence of electronic structures. The calculations were done for nonmagnetic PtBi$_2$ with SOC. We used the maximally localized Wannier functions (MLWFs) to construct a tight-binding model by fitting the DFT band structure[62], where 72 MLWFs were included. The spin polarization was calculated using this tight-binding model. For all calculations, we used the experimentally determined crystal structure (space group $P31m$, $a = b = 6.5705$ Å, $c = 6.1707$ Å).

## Data availability
The data that support the findings of this study are available from the corresponding authors upon reasonable request.

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

## Acknowledgements

The authors would like to thank Darren Peets and Hongxin Yang for helpful discussion. This work is supported by the National Key Research and Development Program of China (Nos. 2017YFA0303600, 2016YFA0300600, and 2017YFA0302901), the National Natural Science Foundation of China (Grant Nos. 11674367,11227902,11927807, U1632266, and 11774399), the Natural Science Foundation of Zhejiang, China (LZ18A040002), the Ningbo Science and Technology Bureau (Grant No. 2018B10060), the German Science Foundation through DFG-SFB 1170 "TOCOTRONICS" (project B04). We acknowledge further financial support from the DFG through the Würzburg–Dresden Cluster of Excellence on Complexity and Topology in Quantum Matter—ct.qmat (EXC 2147, project-id 39085490). S.L.H. would like also to acknowledge the Ningbo 3315 program. We thank the Hiroshima Synchrotron Radiation Center for providing beamtime (project nos. 18AU013, 18AU017, and 18AG018).

## Author contributions

S.L.H., S.Z.X., X.X.W., Y.F., and S.Q. conceived the project. M.Y. and Y.G.S. grew the samples. Y.F. and S.Z.X. carried out the ARPES measurements with the assistance from B.J.F., Q.J., T.X., W.J.L., X.F.Y., M.A., E.F.S. and K.S. S.Q. led the spin-resolved ARPES measurements with the assistance from Q.J., T.X., and W.J.L. X.X.W., H.O.J. and R.T. performed the theoretical calculations. S.L.H., Y.F., S.Z.X., X.X.W. and S.Q. analyzed the data and wrote the paper. All authors discussed the results and commented on the manuscript.

## Competing interests

The authors declare no competing interests.
