## [Peer Review File · Nature Communications]

Reviewer #1 (Remarks to the Author):

The authors reported the characterization of a Rashba splitting state in a very well known material PtBi₂. I think the paper is interesting and well written but to be honest, I do not share the excitement of the authors. I will not call this state as giant as it is larger (but not extremely larger) than BiTeI or recent organic perovskites. The methods used for the characterization are now standard and there is nothing new in that respect. The theory is also Ok though my main concern is in relation to the use of the experimental lattice parameter for the reported results. Any theoretician would worry that they are using a geometry that does not correspond to the theoretical ground state. For example in surface states in STO, this consideration makes a difference. As many of these materials are quite sensitive to strain, it is hard to conclude that theory is capturing correctly the physics. What are the changes in the band structure and the parameters when the real ground state is used? is the "reported giant value" smaller?

I understand that some of the experiments are hard but for me, but fig 3 and fig 5 take the results too far. In the case of Fig 3, the spin-resolved image can be interpreted as the authors suggest but with a large imagination. I would not be able to make an assessment based on that figure, as the quality is very diffuse. A similar situation if Fig 5, c and d. As the used lines are so strong, they guide the reader to the "right result". I would not be able to make the same analysis without the lines. Even more, if the theoretical band structure is obtained with the "wrong" cell parameter but they agree with experiment, what does it say about the theory?... Umm... again, these facts make me think that this work does not have the required threshold of nature communications.

One additional comment is related to the reduced references. I understand is a complex and lively field but recently there are some other claims of Rashba splitting as in BiSb, organic perovskites, PbAu binary alloys, quasi one dimensional Bi, etc. The citation to other materials and how this material is superior is necessary.

Reviewer #2 (Remarks to the Author):

The manuscript describes ARPES and S-ARPES data of in-situ cleaved PtBi₂ in the P31m phase, which reveal a Rashba/Dresselhaus-type spin splitting of bands at the M –point with anisotropic three-dimensional dispersion. The experimental data is in reasonable agreement with DFT and tight-binding calculations of the band structure. The manuscript is mostly well written. Unfortunately, the story is embedded in a number of incorrect claims, likely for advertisement reasons. Moreover, at a few points, a more precise description of the data is required. Both must be corrected prior to publication.

Disregarding the wrong claims, I still believe that the finding of a strongly anisotropic bulk Rashba/Dresselhaus-type band that is centered at a low symmetry point of the BZ is of broad scientific interest, since, to my knowledge, this is a novel type of spin-orbit induced band structure, at least, experimentally. Since spin-orbit effects on band structure are currently in the focus of multiple research areas, e.g., with respect to spinorbitronics, topological insulators, exfoliable 2D materials, and Majorana excitations, this novel type of band structure might trigger new ideas for its exploitation. Hence, if the authors could provide a more decent and correct description of their achievements and can give a more clear-cut possible perspective, I might support publication in Nature Communications.

In detail:

a) The authors claim to provide the first 3D band structure of a Rashba/Dresselhaus-type band, i.e. a band with helical spin splitting and 3D dispersion. However, this has been found previously for the Rashba-split bulk bands of GeTe (e.g. Fig. 3 a-f of Adv. Mat. 28, 560 (16), Fig.2 of J. Phys. Chem. Sol. <https://doi.org/10.1016/j.jpcs.2017.11.010>). These Rashba bands show dispersion along k_z and also an in-plane anisotropy of the dispersion (Fig. 2 e,f B1/B2-bands Adv. Mat. 28, 560 (16)) such that also the anisotropy of the dispersion found for PtBi₂ is not novel.

b) The authors claim that their deduced Rashba coefficient $\alpha_R = 4.36 \text{ eV\AA}$ is larger than the “largest in previous publications”. They attribute the previous record to BiTeI. Both is wrong. To name a few larger values of α_R with respect to BiTeI and PtBi₂: Bi/InAs(110) has been found with 5.5 eV\AA (PRB 98, 075431 (18)), PtSe₂ on Pt(111) exhibits $\sim 6.4 \text{ eV}$ as can be deduced from Fig. 2d (Nat. Commun. 814, 14216 (17)), GeTe provides $\sim 5 \text{ eV}$ as found theoretically in Adv. Mat. 25, 509 (13) and was experimentally confirmed in Adv. Mat. 28, 560 (16), PRB 94, 205111 (16), PRB 94, 201403 (16), Nat. Commun 7, 13071 (17), etc.

Hence, the main claims of the manuscript have to be modified. I firstly propose a more adequate title reflecting the real novelty reading, e.g.: Giant Rashba-Dresselhaus-type spin splitting at the low symmetry M points of PtBi₂ with anisotropic 3D dispersion. Then, I would concentrate on the major novel finding that the symmetry of the M point reveals a cooperative action of Rashba and Dresselhaus terms leading to a strong in-plane anisotropy of the spin splitting as well as to a splitting along k_z .

Further important points to be changed prior to publication:

a) The authors must describe how they determine $k_z = 0$ and the other k_z values (Fig. 5e) in the experiment. Do they assume an inner potential ?

b) The authors claim “good agreement” between Fig. 1(d) and (e), which is wrong. The central star does not appear in experiment, the sizes of the triangles are markedly different in (d) and (e) and the ellipse is barely discernable in the experimental data. Hence, “reasonable agreement” is more adequate

c) The authors claim a Rashba coefficient of 4.36 eV\AA implying a precision of $\sim 0.2 \%$. This is wrong regarding the visible error bars. From visual inspection of Fig. 2a (c-k), I deduce 10-20 meV error bar in ER and $\sim 0.01 \text{ \AA}^{-1}$ in k_0 leading to $\sim 20 \%$ error in αR . The authors must perform a reasonable error estimate and must give reliable numbers afterwards.

d) Seven lines later, the description and visual inspection of Fig. 2c-k result in a different ER of about 80 meV instead of 98 meV. Finally, the band adaption via TB described below eq. (1) implies $\alpha R \approx 2 \text{ eV\AA}$. The authors should bring these markedly different values to a common footing.

e) “... match well ...” is claimed for Fig. 5(e) and (f) albeit the minimum in band dispersion of (f) is not found in the experimental data in (e). This discrepancy should be named and possible reasons for it should be outlined.

f) The two sentences at the end of the manuscript “The multi- ... instabilities.” are not understandable to me and do not contain any reference. They must either be skipped or outlined in a way, that one can understand what is anticipated. If they are skipped, another clear-cut perspective of the data must be given.

g) The check if the helicity in the experiment matches the helicity in the calculation is missing (Is the inner band in the experiment also clockwise?).

i) The calculation finds 90 % spin polarization in Fig. 3d, while the experiment finds only 20 % in Fig. 3b. The difference must be discussed quantitatively in terms of resolution etc.

j) It would help, if the expected diffraction spot angles and intensities for P3 and P31m structure are added to Fig. S1a. The reader should be able to judge the correctness of the assignment.

k) The claimed good agreement between Fig. S5(c) and the DFT results is not shown. Please add the DFT equi-energy contours to Fig. S5.

Minor:

a) The authors write the “materials realizations for spintronics applications are based on spinorbit coupling [1,2]”. However, so far, there are no such applications. Accordingly, ref. [1] writes “The emergent characteristics of these SOC-induced phenomena, which are robust at room temperature, offer several potential applications.” Inserting the “potential” or a “possible” into the actual manuscript would make its intro more correct.

b) The authors write “Rashba splitting was first directly observed ... Au(111) by ... (SpinARPES)”. This is misleading, since “directly” is not precisely defined. E-field dependent SdH beating could be regarded as direct as well. “Rashba splitting was first observed by spinARPES in the Shockley state ...” would make the sentence correct.

- c) I would appreciate, if m_x etc would be also given in their usual units being m_e , i.e $m_x \approx 0.2 m_e$, Experimentalists know these values by heart.
- d) The term “3D map of the Rashba –like ... “ irritates, since the 2D dispersion is displayed and 3D dispersion plays a major role in the claims. I would prefer concreteness: “...show a 3D $E(k_x, k_y)$ map ... and the corresponding electronic structure calculated ...”. The same applies for the caption of Fig. 4.
- e) It should be stated if the scale in Fig. 1d, ... is linear and if not, what is chosen as a scaling.
- f) The wave vector k is partly given as capital letter and partly not. Please adapt.
- g) I believe that Table 2 and 3 result from XRD data, but I am not sure, not being XRD expert. Please clarify in the captions. Moreover, WPa and U_{eq} must be defined somewhere and units should be added to table 2 (I guess Å and Å²).
- h) Photon energy should be added to the captions of Fig. 3a, 4a, S2c.
- i) Scale bar in Fig. S2b should be labeled “spin polarization”.
- j) Typos: page 6 main text: “crossing pint”, page 4 SI: “Dresselhuase”, caption Fig. S4 “Åand” w/o blank.

Reviewer #3 (Remarks to the Author):

I have read the manuscript "Giant anisotropic Rashba like spin splitting in PtBi₂" by Ya Feng and collaborators submitted to Nature Communications for consideration. The main finding reported in this manuscript is a large 3D spin orbit splitting at M points of the Brillouin zone in the P31m phase of single crystal PtBi₂. Using spin resolved ARPES, a characteristic helical spin structure is observed in elliptic contours evidencing significant anisotropy in the in plane spin splitting. Cooperative Rashba and Dresselhaus spin orbit interactions are invoked to explain the spin splitting anisotropy which, as a result, turns out to be 3D in nature as discussed by theoretical modelling. 3D character of the spin splitting is confirmed by photon energy dependent ARPES experiments which show significant k_z dispersion.

The paper contains a set of good quality data demonstrating the large anisotropic Rashba splitting in PtBi₂ single crystals, a compound which has recently raised interest for interesting transport properties and the possibility of topological phases (arxiv 1809. 06507, PRL 118 256601 (2017), Nat Comms. 9 3249 (2018)). In this sense the present manuscript showing evidence for a very strong non-conventional (3D) Rashba splitting is very timely. The 3D Rashba interaction manifesting through cooperative Rashba – Dresselhaus interaction is an interesting new scenario and it is different to the 2D splitting reported in the BiTeI linked to Rashba interaction in surface planes. As such, I think that this manuscript should be published as it may contribute to the understanding of highly exotic novel behaviors in this material.

My only comment concerns the somewhat poorer quality of the data showing the k_z dispersion of Figure 5. In particular energy momentum curves at 11 eV shows a highly suppressed (right) Rashba splitted band. Could authors comment on the origin of these features? Is interpretation robust towards them?

Reviewer #3 (Remarks to the Author):

I have read the manuscript "Giant anisotropic Rashba like spin splitting in PtBi₂" by Ya Feng and collaborators submitted to Nature Communications for consideration. The main finding reported in this manuscript is a large 3D spin orbit splitting at M points of the Brillouin zone in the P31m phase of single crystal PtBi₂. Using spin resolved ARPES, a characteristic helical spin structure is observed in elliptic contours evidencing significant anisotropy in the in plane spin splitting. Cooperative Rashba and Dresselhaus spin orbit interactions are invoked to explain the spin splitting anisotropy which, as a result, turns out to be 3D in nature as discussed by theoretical modelling. 3D character of the spin splitting is confirmed by photon energy dependent ARPES experiments which show significant k_z dispersion.

The paper contains a set of good quality data demonstrating the large anisotropic Rashba splitting in PtBi₂ single crystals, a compound which has recently raised interest for interesting transport properties and the possibility of topological phases (arxiv 1809. 06507, PRL 118 256601 (2017),

Nat Comms. 9 3249 (2018)). In this sense the present manuscript showing evidence for a very strong non-conventional (3D) Rashba splitting is very timely. The 3D Rashba interaction manifesting through cooperative Rashba – Dresselhaus interaction is an interesting new scenario and it is different to the 2D splitting reported in the BiTeI linked to Rashba interaction in surface planes. As such, I think that this manuscript should be published as it may contribute to the understanding of highly exotic novel behaviors in this material.

My only comment concerns the somewhat poorer quality of the data showing the k_z dispersion of Figure 5. In particular energy momentum curves at 11 eV shows a highly suppressed (right) Rashba splitted band. Could authors comment on the origin of these features? Is interpretation robust towards them?

Response to Referee 1

We thank the referee for reviewing our manuscript and offering his/her helpful comments. We also appreciate the referee for the encouraging comments on our work "*I think the paper is interesting and well written...*". Please find our detailed reply to the comments and questions below.

C1: The authors reported the characterization of a Rashba spitting state in a very well known material PtBi₂. I think the paper is interesting and well written but to be honest, I do not share the excitement of the authors. I will not call this state as giant as it is larger (but not extremely larger) than BiTeI or recent organic perovskites.

Reply: We thank the referee for reminding us that we should use the word "giant" carefully. Among the reported experimental results of Rashba splitting, the α_R ($=4.36\text{eV}\text{\AA}$) value in PtBi₂ is indeed not the largest one. (It is larger than BiTeI and still one of the largest values of bulk Rashba splitting). However, the main novelty of our work is the experimental discovery of 3D Rashba-like spin splitting in PtBi₂, which constitutes a unconventional Rashba-like splitting.

While "giant" is often used in the literature to describe splitting of similar magnitude, we have replaced "giant" with "large" in three sentences in the interests of precision:

Sentence 1: "In particular, the **giant** spin splitting emerges at the M points of the Brillouin zone instead of the Γ point, where Rashba type splitting is usually found." was replaced with "In particular, the **large** spin splitting emerges at the M points of the Brillouin zone instead of the Γ point, where Rashba type splitting is usually found."

Sentence 2: "First, the reduced point group symmetry at M along with the bulk inversion asymmetry allow the Dresselhaus- and Rashba-type SOC terms to cooperatively yield a **giant** anisotropic spin splitting with a helical texture around the M point." was replaced with "The reduced point group symmetry at M along with the bulk inversion asymmetry allow the Dresselhaus- and Rashba-type SOC terms to cooperatively yield a **large** anisotropic spin splitting with a helical texture around the M point."

Sentence 3: "It is evident that the bulk states exhibit **giant** Rashba-type splitting with the crossing point at about $E_b = 520$ meV. " was replaced with "It is evident that the bulk states exhibit **large** Rashba-type splitting with the crossing point at about $E_b = 520$ meV. "

C2: The methods used for the characterization are now standard and there is nothing new in that respect.

Reply: This is not completely correct, although it's not crucial to the novelty of the results. The spin-resolved ARPES used in this work employed the latest state-of-art image-type VLEED-type spin detector[PRL 116, 177601 (2016)]. This newly developed machine can obtain 2D spin-resolved band mapping simultaneously, while the traditional spin detectors

can only collect a 0D spin-resolved data at a time.

In this sense, the experimental method does have something new and interesting to offer. We should like to mention the method itself is not crucial to the novelty of the 3D Rashba-like spin splitting reported for the first time in this manuscript.

C3: *The theory is also Ok though my main concern is in relation to the use of the experimental lattice parameter for the reported results. Any theoretician would worry that they are using a geometry that does not correspond to the theoretical ground state. For example in surfaces states in STO, this consideration makes a difference. As many of these materials are quite sensitive to strain, it is hard to conclude that theory is capturing correctly the physics. What are the changes in the band structure and the parameters when the real ground state is used? is the "reported giant value" smaller?*

Reply: We thank the referee for this comment and understand the referee's concern. In first-principle calculations, it is well known that LDA underestimates volumes of crystal while GGA overestimates those. Therefore, to assess the accuracy of DFT calculations, we should compare the theoretical lattice parameters and experimental ones. For bulk systems, their lattice parameters can be determined by XRD measurements. Usually theoretical calculations using experimental lattice parameters can reproduce the electronic structure (ARPES measurements) better than the relaxed parameters. A well-known example is the iron based superconductors. DFT calculations tend to underestimate the Fe-As bond in the relaxation and calculations with experimental lattice parameters are much better. (see Phys. Rev. B 78, 094511 (2008), Phys. Rev. Lett. 101, 047001 (2008), Phys. Rev. B 78, 085104 (2008))

For PtBi₂ crystal, the lattice parameters used for calculations are obtained from single crystal XRD measurements. Then we further minimize the total energy allowing lattice relaxation, and obtained lattice constants are $a=6.678 \text{ \AA}$ and $c=6.316 \text{ \AA}$ ($a=6.5705 \text{ \AA}$ and $c=6.1707 \text{ \AA}$ in experiments). Indeed we find the overestimation of crystal volume within GGA functional and the obtained band structure is shown in Fig. R1, in comparison with bands from experimental parameters. While the Rashba bands calculated with the relaxed lattice parameters shift towards the Fermi level, the Rashba splitting is almost the same as the one with experimental lattice parameters.

Fig. R1: The comparison of calculated band structure by using experimental (red line) and relaxed (blue line) lattice parameters.

In summary: 1. Both the calculated results using experimental and relaxed lattice parameter

can capture the observed Rashba band dispersion and the former shows better agreement with the measured band structures; 2. Both the relaxed and experimental lattice parameter yield a similar strength of Rashba splitting.

C4: *I understand that some of the experiments are hard but for me, but fig 3 and fig 5 take the results too far. In the case of Fig 3, the spin-resolved image can be interpreted as the authors suggest but with a large imagination. I would not be able to make an assessment based on that figure, as the quality is very diffuse. A similar situation if Fig 5, c and d. As the used lines are so strong, they guide the reader to the "right result". I would not be able to make the same analysis without the lines.*

Reply: We thank the referee for these helpful comments and the opportunity to make our presentation more convincing.

Fig. R2: (a): the previous data taken in sweep mode; (b): the new data collected in fixed mode.

For Fig. 3, in order to improve the data quality of the spin-resolved data, we remeasured the spin polarization of the band using fixed mode, which is time-consuming but has higher resolution than the sweep mode used for the previous Fig. 3. The newly obtained spin-resolved image is compared against the previous data in Fig. R2, showing greatly improved resolution. For a spin-resolved ARPES measurement, such data quality in a 2D plot is very impressive. Prior to the advent of this 2D detector, the quality of the spin-resolved MDCs in Fig. 3b would have been considered more than sufficient to establish our conclusion.

We have replaced the data in Fig. 3 (a) and (b) with the newly measured data.

In Fig 5, we made the guide lines weaker, in order to show the band dispersion more clearly. We also performed second-derivative data processing along the energy direction for Fig 5(a-d), as shown in the supplementary as Fig. S6. As shown in Fig. R3, the band dispersion is more clearly visible compared to the original image.

The new results in Fig. 3 and reprocessed data in Fig. 5 and Fig. S6 now support our conclusion more convincingly.

Fig.R3: The original data in Fig.5(c) and (d) without lines (left) and the corresponding second derivative data (right).

C5: *Even more, if the theoretical band structure is obtained with the "wrong" cell parameter but they agree with experiment, what does it say about the theory?... Umm... again, these facts make me think that this work does not have the required threshold of nature communications.*

Reply: We have addressed the lattice parameter above, and also note that the calculated band structures show reasonable agreement with the experiment results.

C6: *One additional comment is related to the reduced references. I understand is a complex and lively field but recently there are some other claims of Rashba splitting as in BiSb, organic perovskites, PbAu binary alloys, quasi one dimensional Bi, etc. The citation to other materials and how this material is superior is necessary.*

Reply: We thank the referee for the helpful suggestion. We have added some references as mentioned by the referee and revised the third and fourth paragraph in the introduction.

Response to Referee 2

The manuscript describes ARPES and S-ARPES data of in-situ cleaved PtBi₂ in the P31m phase, which reveal a Rashba/Dresselhaus-type spin splitting of bands at the M –point with anisotropic three-dimensional dispersion. The experimental data is in reasonable agreement with DFT and tight-binding calculations of the band structure. The manuscript is mostly well written. Unfortunately, the story is embedded in a number of incorrect claims, likely for advertisement reasons. Moreover, at a few points, a more precise description of the data is required. Both must be corrected prior to publication.

Reply: We thank the referee for helping us describe our data and its context more precisely. The detailed reply and related modifications can be found below.

Disregarding the wrong claims, I still believe that the finding of a strongly anisotropic bulk Rashba/Dresselhaus-type band that is centered at a low symmetry point of the BZ is of broad scientific interest, since, to my knowledge, this is a novel type of spin-orbit induced band structure, at least, experimentally. Since spin-orbit effects on band structure are currently in the focus of multiple research areas, e.g., with respect to spinorbitronics, topological insulators, exfoliable 2D materials, and Majorana excitations, this novel type of band structure might trigger new ideas for its exploitation. Hence, if the authors could provide a more decent and correct description of their achievements and can give a more clear-cut possible perspective, I might support publication in Nature Communications.

Reply: We thank the referee for the positive evaluation of our work “...I still believe that the finding of a strongly anisotropic bulk Rashba/Dresselhaus-type band that is centered at a low symmetry point of the BZ is of broad scientific interest, ... this is a novel type of spin-orbit induced band structure, at least, experimentally.”

The referee's detailed suggestions were very valuable for improving our manuscript.

C1:a) *The authors claim to provide the first 3D band structure of a Rashba/Dresselhaus-type band, i.e. a band with helical spin splitting and 3D dispersion. However, this has been found previously for the Rashba-split bulk bands of GeTe (e.g. Fig. 3 a-f of Adv. Mat. 28, 560 (16), Fig.2 of J. Phys. Chem. Sol. <https://doi.org/10.1016/j.jpccs.2017.11.010>). These Rashba bands show dispersion along k_z and also an in-plane anisotropy of the dispersion (Fig. 2 e,f B1/B2-bands Adv. Mat. 28, 560 (16)) such that also the anisotropy of the dispersion found for PtBi₂ is not novel.*

Reply: We thank the referee for this good comment, which touches on the key point of our work.

First, we need to clarify the difference between "3D Rashba dispersion" and "3D Rashba splitting". While Rashba bands splitting along in-plane momentum directions, there are three cases for Rashba bands along k_z (out-of-plane direction):

1. There is no dispersion along k_z . This is typical case for Rashba splitting at Au(111) surface.
2. The Rashba bands have dispersion along k_z , but they are spin degenerate along k_z for

$k_x=k_y=0$ as sketched in Fig. R4(a). This is the case for Rashba splitting previously reported in bulk state, such as that in BiTeI [Nat. Mater. 10, 521-526 (2011); PRB 86, 085204(2012)]. The previous results on GeTe [Adv. Mat. 28, 560 (16)] and its related material system [J. Phys. Chem. Sol. <https://doi.org/10.1016/j.jpcs.2017.11.010>], as correctly pointed out by the referee, are also belonging to this category.

3. In addition to dispersion along k_z , the Rashba-split bands show band splitting along k_z for $k_x=k_y=0$, as sketched in Fig.R4(b). In this scenario, the Rashba-like bands actually show splitting along k_x , k_y , and k_z in momentum space (we call this "3D Rashba-like spin splitting"). This is experimentally realized here, for the first time, in PtBi₂, and it constitutes the main novelty of this manuscript. We included the sketch in Fig. 5 to clarify the difference between "3D dispersion" and "3D spin splitting".

Fig.R4: Sketch of two kinds of bulk Rashba band structures. **a.** It has dispersion along k_z with spin degenerate along k_z for $k_x=k_y=0$. **b.** In addition to the dispersion along k_z direction, it shows band splitting along k_z .

C2:b) *The authors claim that their deduced Rashba coefficient $\alpha_R = 4.36 \text{ eV\AA}$ is larger than the "largest in previous publications". They attribute the previous record to BiTeI. Both is wrong. To name a few larger values of α_R with respect to BiTeI and PtBi₂: Bi/InAs(110) has been found with 5.5 eV\AA (PRB 98, 075431 (18)), PtSe₂ on Pt(111) exhibits $\sim 6.4 \text{ eV}$ as can be deduced from Fig. 2d (Nat. Commun. 814, 14216 (17)), GeTe provides $\sim 5 \text{ eV}$ as found theoretically in Adv. Mat. 25, 509 (13) and was experimentally confirmed in Adv. Mat. 28, 560 (16), PRB 94, 205111 (16), PRB 94, 201403 (16), Nat. Commun 7, 13071 (17), etc.*

Reply: The referee correctly pointing out that our claim "largest in previous publications" has to be modified. We cited the related references including those mentioned by the referee. And we rephrased the following sentences accordingly:

1. In the sentence "Due to its inversion asymmetry and reduced symmetry at the M point, Rashba-type as well as Dresselhaus-type SOC cooperatively yield a giant 3D spin splitting in PtBi₂, which exceeds those in related material classes.", we removed "which exceeds those in related material classes".

2. The sentence "This value is even larger than that in BiTeI ($\alpha_R = 3.85 \text{ eV\AA}$) [32], which is

the largest in previous publications" was change to "This value is even larger than that in BiTeI ($\alpha_R = 3.85 \text{ eV\AA}$) [32] and is one of the largest **bulk** Rashba state **occurs at the lower symmetry M points among** previous publications".

C3: *Hence, the main claims of the manuscript have to be modified. I firstly propose a more adequate title reflecting the real novelty reading, e.g.:*

Giant Rashba-Dresselhaus-type spin splitting at the low symmetry M points of PtBi₂ with anisotropic 3D dispersion.

Then, I would concentrate on the major novel finding that the symmetry of the M point reveals a cooperative action of Rashba and Dresselhaus terms leading to a strong in-plane anisotropy of the spin splitting as well as to a splitting along k_z .

Reply: We thank the referee for the suggestion to improve our title. The referee suggested that we add the words "3D dispersion". However, as clarified above, there is a significant difference between "3D dispersion" and "3D Rashba splitting", and the main novelty of our work is that we observe Rashba-like splitting along all three momentum directions (k_x, k_y, k_z), not merely 3D dispersion.

However, to avoid any confusion by using the phrase "3D Rashba-like spin splitting" in the title, we use the words "Rashba-like spin splitting along three momentum directions". Finally, we replaced "Giant 3D anisotropic Rashba-like spin splitting in PtBi₂" with "Giant anisotropic Rashba-like spin splitting along three momentum directions in PtBi₂".

C4: *a) The authors must describe how they determine $k_z = 0$ and the other k_z values (Fig. 5e) in the experiment. Do they assume an inner potential ?*

Reply: These details should indeed be available. Accordingly, we have added a paragraph to the Supplementary describing how we identified $k_z=0$.

C5: *b) The authors claim "good agreement" between Fig. 1(d) and (e), which is wrong. The central star does not appear in experiment, the sizes of the triangles are markedly different in (d) and (e) and the ellipse is barely discernable in the experimental data. Hence, "reasonable agreement" is more adequate*

Reply: We thank the referee for the valuable suggestion. We have changed "good agreement" to "reasonable agreement".

C6: *c) The authors claim a Rashba coefficient of 4.36 eV\AA implying a precision of $\sim 0.2 \%$. This is wrong regarding the visible error bars. From visual inspection of Fig. 2a (c-k), I deduce $10\text{-}20 \text{ meV}$ error bar in E_R and $\sim 0.01 \text{ \AA}^{-1}$ in k_0 leading to $\sim 20 \%$ error in α_R . The authors must perform a reasonable error estimate and must give reliable numbers afterwards.*

Reply: We thank the referee for the suggestion. We describe our estimation of the uncertainty of α_R below:

Since $\alpha_R = 2E_R/k_0$, $\Delta\alpha_R/\alpha_R = [(\Delta E_R/E_R)^2 + (\Delta k_0/k_0)^2]^{1/2}$. The uncertainty in the MDC fitting to

determine k_0 is $\Delta k_0 = 0.001 \text{ \AA}^{-1}$, and the uncertainty in the EDC fitting to determine E_R is $\Delta E_R = 2 \text{ meV}$. Combined with $k_0 = 0.045 \text{ \AA}^{-1}$ and $E_R = 98 \text{ meV}$, we obtain $\Delta \alpha_R / \alpha_R \approx 3.1\%$, $\alpha_R = 4.36 \pm 0.14 \text{ eV \AA}$.

In the main text, we added this uncertainty.

C7: *d) Seven lines later, the description and visual inspection of Fig. 2c-k result in a different E_R of about 80 meV instead of 98 meV. Finally, the band adaption via TB described below eq. (1) implies $\alpha_R \approx 2 \text{ eV \AA}$. The authors should bring these markedly different values to a common footing.*

Reply: We thank the referee for checking our manuscript very carefully. There was a typo in Fig. 2k. The image in Fig. 2k should be labeled as $E_b = 620 \text{ meV}$ instead of 600 meV. This is consistent with Fig. 2b, where the bottom of the Rashba band is located at 620 meV. In Fig. R5, the images taken at $E_b = 580 \text{ meV}$, $E_b = 600 \text{ meV}$ and $E_b = 620 \text{ meV}$ are compared to show the difference. A simple visual inspection of Fig. 2c-k now gives an E_R of about 100 meV, which is consistent with the former value of 98 meV.

The discrepancy in α_R arises from a disagreement between the calculated and measured band structure. Here we used an effective model to analyze the origin of anisotropic splitting in experiments. Since the Rashba energy is underestimated in the DFT calculations, the value of α_R is also underestimated in theoretical calculations. The quantitative difference between the theoretical calculations and experiments will not affect our conclusions.

Fig.R5: The original Fig. 2b (left) and constant energy contours at $E_b = 580 \text{ meV}$, 600 meV and 620 meV respectively (right).

C8: *e) "... match well ..." is claimed for Fig. 5(e) and (f) albeit the minimum in band dispersion of (f) is not found in the experimental data in (e). This discrepancy should be named and possible reasons for it should be outlined.*

Reply: We thank the referee for the good suggestion. Fig. 5e was made by extracting energy distribution curves at the M point taken at different photon energies. For ARPES measurements, the band dispersion along the k_z direction obtained by varying incident photon energy has much poorer momentum resolution than the in-plane momentum resolution, which is mainly related to the angular resolution in tilting the samples.

In the revised paper we changed the sentence "The obtained band splitting and dispersion match well with the calculated band dispersion along $M-L$ direction as shown in Fig. 5(f)." to "The obtained band splitting along k_z direction can be qualitatively captured by the calculated

results as shown in Fig. 5(f). The specific shape of the dispersions is a little different, which may be attributed to the poor momentum resolution along the out-of-plane direction in ARPES measurements.”

C9: *f) The two sentences at the end of the manuscript “The multi- ... instabilities.” are not understandable to me and do not contain any reference. They must either be skipped or outlined in a way, that one can understand what is anticipated. If they are skipped, another clear-cut perspective of the data must be given.*

Reply: We remove these three sentences in conclusion and add a new sentence to speculate further research for exploring the new 3D Rashba physics and materials for potential applications in spintronics.

C10: *g) The check if the helicity in the experiment matches the helicity in the calculation is missing (Is the inner band in the experiment also clockwise?).*

Reply: From the data in Fig. 3, we can get the spin structure as shown in Fig.R6. The lower part shows the spin structure of the split bands. We label the spin direction at the corresponding position on the constant energy surface as shown on the upper part. The inner band has clockwise spin polarization as calculated.

Fig.R6 Spin structure of the splitting bands.

C11: *i) The calculation finds 90 % spin polarization in Fig. 3d, while the experiment finds only 20 % in Fig. 3b. The difference must be discussed quantitatively in terms of resolution etc.*

Reply: In the real spin-resolved photoemission measurements, there are many facts can make the spin polarization smaller than the calculated value. These facts mainly come from: 1. Impurities and defects at the cleaved surface of samples; 2. The beam size of incident light is usually of 0.5-2 mm, and the measured spin polarization is averaged over this area given the inhomogeneity of the samples. 3. The resolution as well as the inelastic scattering of the photoelectrons during the photoemission measurements will also increase the unpolarized background and lower the measured spin polarization.

C12: j) *It would help, if the expected diffraction spot angles and intensities for $P3$ and $P31m$ structure are added to Fig. S1a. The reader should be able to judge the correctness of the assignment.*

Reply: Fig. S1a does not show the single-crystal diffraction data used to refine the crystal structure, and cannot distinguish between the two structures. Simulated XRD powder patterns for the $P31m$ and $P\bar{3}$ structures are shown in Fig. R7. These structures can be distinguished by the intensity ratios among the peaks, but not by peak positions.

The normal process of single crystal XRD analysis is to use the R values from refinements in different space groups to judge which space group the sample belongs to. However, refinements in $P\bar{3}$ were not successful, likely indicating that the intensity ratios are completely incompatible with $P\bar{3}$. This prevents a direct comparison of how well each space group describes the experimental data, but gives us confidence that our space group is correct.

Fig.R7 Simulated XRD powder patterns for $PtBi_2$ with $P31m$ and $P\bar{3}$ structures.

C13:k) *The claimed good agreement between Fig. S5(c) and the DFT results is not shown. Please add the DFT equi-energy contours to Fig. S5.*

Reply: We thank the referee for the suggestion. The DFT equi-energy contour has been added to Fig. S5 as Fig. S5d, and we changed “good agreement” to “reasonable agreement”.

Reply to the referee's comments marked as minor:

C14: a) *The authors write the “materials realizations for spintronics applications are based on spinorbit coupling [1,2]”. However, so far, there are no such applications. Accordingly, ref. [1] writes “The emergent characteristics of these SOC-induced phenomena, which are robust at room temperature, offer several potential applications.” Inserting the “potential” or a “possible” into the actual manuscript would make its intro more correct.*

Reply: We have inserted the word “potential” in the sentence “The present material

realizations for spintronics applications are based on spin-orbit coupling (SOC)”.

C15:b) *The authors write “Rashba splitting was first directly observed ... Au(111) by ... (Spin ARPES)”. This is misleading, since “directly” is not precisely defined. E-field dependent SdH beating could be regarded as direct as well. “Rashba splitting was first observed by spin ARPES in the Shockley state ...” would make the sentence correct.*

Reply: We thank the referee for offering this precise statement. We changed the sentence “... was first directly observed in the Shockley surface state of Au(111) by (spin- and) angle-resolved photoemission spectroscopy (ARPES)” to “...was first observed by spin- and angle-resolved photoemission spectroscopy (Spin-ARPES) in the Shockley surface state of Au(111)”.

C16:c) *I would appreciate, if m_x etc would be also given in their usual units being m_e , i.e $m_x \approx 0.2m_e$, Experimentalists know these values by heart.*

Reply: We give the value of m in units of m_e . “ $m_x=0.17m_e$, $m_y=0.85m_e$ ” has been added in main text.

C17: d) *The term “3D map of the Rashba –like ... “ irritates, since the 2D dispersion is displayed and 3D dispersion plays a major role in the claims. I would prefer concreteness: “...show a3D $E(k_x,k_y)$ map ... and the corresponding electronic structure calculated ...”. The same applies for the caption of Fig. 4.*

Reply: We changed “3D map” to “3D $E(k_x,k_y)$ map” in the sentence “We also show a 3D map of the Rashba-like spin split bands as determined by ARPES in Fig. 4(a) ...”. We changed the caption of Fig. 4a from “Three-dimensional constant energy-contours of ARPES data” to “3D $E(k_x,k_y)$ map from ARPES data”.

C18:e) *It should be stated if the scale in Fig. 1d, ...is linear and if not, what is chosen as a scaling.*

Reply: The scale() is linear.

C19: f) *The wave vector \mathbf{k} is partly given as capital letter and partly not. Please adapt.*

Reply: We thank the referee for his/her careful check. We changed the capital letter “K” to “k” in Fig.1, 2, 4, 5, and S2.

C20:g) *I believe that Table 2 and 3 result from XRD data, but I am not sure, not being XRD expert. Please clarify in the captions. Moreover, WPa and U_{eq} must be defined somewhere and units should be added to table 2 (I guess Å and Å^2).*

Reply: Yes, Tables 2 and 3 result from XRD data. We clarify that by adding “result from single-crystal XRD analysis” to their captions. WP and Ueq have been defined, and the unit is also added to Table 2. WP means Wyckoff position which is used to denote the symmetry of equivalent atoms in a cell.

C21:h) *Photon energy should be added to the captions of Fig. 3a, 4a, S2c.*

Reply: Thank you very much. We added photon energy to the captions of Fig.3a, 4a, and S2c.

C22:i) *Scale bar in Fig. S2b should be labeled “spin polarization”.*

Reply: We added the sentence “The scale bars in a, c and d represent spin polarization” in the end of the caption of Fig. 3, and added the sentence “the scale bar represents spin polarization” in the caption of Fig. S2b.

C23:j) *Typos: page 6 main text: “crossing pint”, page 4 SI: “Dresselhause”, caption Fig. S4 “Å and” w/o blank.*

Reply: We corrected “crossing pint” to “crossing point” and “Dresselhause” to “Dresselhaus”, and added a blank before the word “and” in the caption to Fig.S4.

Response to Referee 3

I have read the manuscript “Giant anisotropic Rashba like spin splitting in PtBi2” by Ya Feng and collaborators submitted to Nature Communications for consideration. The main finding reported in this manuscript is a large 3D spin orbit splitting at M points of the Brillouin zone in the P31m phase of single crystal PtBi2. Using spin resolved ARPES, a characteristic helical spin structure is observed in elliptic contours evidencing significant anisotropy in the in plane spin splitting. Cooperative Rashba and Dresselhaus spin orbit interactions are invoked to explain the spin splitting anisotropy which, as a result, turns out to be 3D in nature as discussed by theoretical modelling. 3D character of the spin splitting is confirmed by photon energy dependent ARPES experiments which show significant kz dispersion.

The paper contains a set of good quality data demonstrating the large anisotropic Rashba splitting in PtBi2 single crystals, a compound which has recently raised interest for interesting transport properties and the possibility of topological phases (arxiv 1809. 06507, PRL 118 256601 (2017), Nat Comms. 9 3249 (2018)). In this sense the present manuscript showing evidence for a very strong non-conventional (3D) Rashba splitting is very timely. The 3D Rashba interaction manifesting through cooperative Rashba – Dresselhaus interaction is an interesting new scenario and it is different to the 2D splitting reported in the BiTeI linked to Rashba interaction in surface planes. As such, I think that this manuscript should be published as it may contribute to the understanding of highly exotic novel behaviors in this material.

Reply: We thank the referee for carefully reviewing our manuscript. The referee correctly captured the main novelty of our work in his/her comment “*In this sense the present manuscript showing evidence for a very strong non-conventional (3D) Rashba splitting is very timely.* We also appreciate the referee's positive comments: “*I think that this manuscript should be published as it may contribute to the understanding of highly exotic novel behaviors in this material*”.

Please find our detailed reply to the referee's question below:

C1:*My only comment concerns the somewhat poorer quality of the data showing the kz dispersion of Figure 5. In particular energy momentum curves at 11 eV shows a highly suppressed (right) Rashba splitted band. Could authors comment on the origin of these features? Is interpretation robust towards them?*

Reply: We thank the referee for the comments. We show the original data of Fig. 5 without guidelines in Fig. R8. The spin-split bands are visible without the guides. In order to show these bands more clearly, we performed second derivative data processing along the energy direction, and the results are shown in the Supplementary as Fig. S6.

The slightly suppressed (right) Rashba-split band at 11eV is likely caused by matrix element

effects, which are quite common in ARPES measurements (we show an example of ARPES measurement of Bi_2Se_3 below.)

The dispersion along k_z in Fig. 5e was obtained by extracting energy distribution curves at the M point taken at different photon energies. In ARPES measurements, the band dispersion along the k_z direction obtained by varying incident photon energy has much poor momentum resolution than the in-plane momentum resolution, which is mainly related to the angular resolution in tilting the samples.

Fig. R8:a-d, Energy-momentum image mapped by ARPES taken at different photon energies.

Example of matrix element effects in ARPES:

(Image from Figure 3 of *Nat. Commun.* 5, 3382 (2014))

Band structure of the Bi_2Se_3 topological insulator measured by ARPES. The right branch of the upper Dirac cone and the left branch of the lower Dirac cone are suppressed.

Summary of changes:

For figures:

1. We updated Fig.1. We changed “K” to “k” in **d** and **e**.
2. We updated Fig.2. We corrected the label of Fig. 2k “Eb = 600meV” to “Eb = 620meV”. We changed “K” to “k”.
3. We updated Fig. 3a and 3b with new data. We added “($h\nu = 8.4$ eV)”, “The scale bars in **a**, **c** and **d** represent spin polarization.” in the caption.
4. We updated Fig. 4a by changing the energy scale from -0.6eV to -0.62eV and the corresponding constant energy contours. We changed “K” to “k”. In the caption, we changed the sentence “Three-dimensional constant energy-contours of ARPES data.” to “3D $E(\mathbf{k}_x, \mathbf{k}_y)$ map from ARPES data ($h\nu = 9$ eV).”
5. We updated Fig. 5a-d by changing the color of guidelines from dark blue to light blue. We changed “K” to “k”.
6. We updated Fig. 5f and added Fig. 5g to clarify the difference between "3D dispersion" and "3D spin splitting".
7. For the caption of Fig. 5: “the M–L direction as sketched in the three-dimensional Brillouin zone shown in the inset” was added to caption of Fig. 5e; the caption of Fig. 5f was changed to “Calculated band structure of PtBi₂ along in-plane and out-of-plane momentum directions. The red and blue solid lines show the in-plane band structures at different k_z . The red and green dashed lines show the band dispersions along k_z direction, indicating Rashba splitting along k_z ; “Sketch of Rashba bands which have dispersion along k_z but remain spin degenerate along k_z for $k_x=k_y=0$ ” was added as the caption of Fig. 5g.
8. We changed “K” to “k” in Fig.S2. We added photon energy to the caption of Fig. S2c. We added the sentence “the scale bar represents spin polarization” in the caption of Fig. S2b
9. We added a blank before the word “and” in the caption on Fig.S4.
10. We added the constant energy contour from DFT calculation to Fig. S5 as Fig. S5d according to the referee’s suggestion. The first words “Fermi surface” were replaced with “constant energy contours”.
11. We added Fig.S6 in the supplementary material. It shows second derivative E-k image of Fig. 5a-d. The band structure can be distinguished more clearly.

For text:

1. We changed the title to “**Giant anisotropic Rashba-like spin splitting along three momentum directions in PtBi₂**”.
2. We deleted the sentence “which exceeds those in related material classes” in the abstract.
3. In the abstract, “We report the discovery of a giant anisotropic 3D Rashba-like spin splitting with a helical spin polarization around the M points” was replaced with “We report the discovery of a giant anisotropic Rashba-like spin splitting along three

momentum dimensions (3D Rashba-like spin splitting) with a helical spin polarization around the M points”

4. In the abstract, “ The experimental realization of 3D Rashba-like spin texture paves the way to the future exploration of a new class of unprecedented material functionalities for spintronics applications.” was replaced with “The experimental realization of 3D Rashba-like spin splitting not only has fundamental interests but also paves the way to the future exploration of a new class of unprecedented material functionalities for spintronics applications.”

5. We replaced the word "giant " with "large" in three sentences to state more precisely.

Sentence 1 : " In particular, the **giant** spin splitting emerges at the M points of the Brillouin zone instead of the Γ point, where Rashba type splitting is usually found." was replaced with

" In particular, the **large** spin splitting emerges at the M points of the Brillouin zone instead of the Γ point, where Rashba type splitting is usually found."

Sentence 2: "First, the reduced point group symmetry at M along with the bulk inversion asymmetry allow the Dresselhaus- and Rashba-type SOC terms to cooperatively yield a **giant** anisotropic spin splitting with a helical texture around the M point." was replaced with " First, the reduced point group symmetry at M along with the bulk inversion asymmetry allow the Dresselhaus- and Rashba-type SOC terms to cooperatively yield a **large** anisotropic spin splitting with a helical texture around the M point."

Sentence 3: "It is evident that the bulk states exhibit **giant** Rashba-type splitting with the crossing point at about $E_b = 520$ meV. " was replaced with " It is evident that the bulk states exhibit **large** Rashba-type splitting with the crossing point at about $E_b = 520$ meV. "

6. We inserted the word “potential” in the second sentence of the *Introduction* “The present material realizations for spintronics applications are based on spin-orbit coupling (SOC)” (page 3)

7. The third paragraph in the introduction was revised:

The sentence “Rashba splitting was first directly observed in the Shockley surface state of Au(111) by (spin- and) angle-resolved photoemission spectroscopy (ARPES)” was replaced with “Rashba splitting was first observed by spin- and angle- resolved photoemission spectroscopy (Spin-ARPES) in the Shockley surface state of Au(111)” (page 3).

The sentences “In particular, BiTeI gained attention for its giant Rashba splitting. However, in all known instances the spin splitting is two-dimensional, with no dispersion in the perpendicular direction in momentum space.” was replaced with “In particular, BiTeI and GeTe gained attention for their giant Rashba splitting in bulk states. However, in all known

instances, the Rashba bands remain spin degenerate along out-of-plane direction k_z for $k_x = k_y = 0$ as sketched in Fig. 5(g), while they show spinsplitting along in-plane momentum directions, namely, k_x and k_y directions.” (page3).

The sentence “While there is no principal symmetry exclusion argument against 3D spin splitting induced by inversion symmetry breaking, ...” was replaced with “While there is no principal symmetry exclusion argument against Rashba spin splitting along three momentum directions (3D Rashba spin splitting) induced by inversion symmetry breaking, ...” (page3).

8. In the fourth paragraph of the introduction:

We deleted the sentence “which is the largest reported one to date[32].”

“First, the reduced point group symmetry at M along with the bulk inversion asymmetry allow the Dresselhaus- and Rashba-type SOC terms to cooperatively yield a large anisotropic spin splitting with a helical texture around the M point. Second, as there are three M points in the trigonal Brillouin zone of our PtBi₂ crystals, there is a three-fold set of SOC-split non-degenerate Fermi pockets centered around the M points.” was replaced with “The reduced point group symmetry at M along with the bulk inversion asymmetry allow the Dresselhaus- and Rashba-type SOC terms to cooperatively yield a large 3D anisotropic spin splitting with a helical texture around the M point. Our observation is therefore not only of fundamental interest for Rashba physics, but also of some potential applications in spintronics.” (page 4).

9. We changed “good agreement” to “reasonable agreement” in the first paragraph of *Results and Discussions* (page 5).

10. We changed “ $k_0 \approx 0.045 \text{ \AA}^{-1}$ ”, “ $E_R \approx 98 \text{ meV}$ ” and “ $\alpha_R \approx 4.36 \text{ eV \AA}$ ” to “ $k_0 = 0.045 \pm 0.001 \text{ \AA}^{-1}$ ”, “ $E_R = 98 \pm 2 \text{ meV}$ ” and “ $\alpha_R = 4.36 \pm 0.14 \text{ eV \AA}$ ”, respectively (in the second paragraph of *Results and Discussions* on page 6).

11. The sentence “This value is even larger than that in BiTeI ($\alpha_R = 3.85 \text{ eV \AA}$)[32], which is the largest in previous publications” was changed to be “ This value is even larger than that in BiTeI ($\alpha_R = 3.85 \text{ eV \AA}$) [32] and is one of the largest bulk Rashba state occurs at the lower symmetry M points among previous publications.” (in the second paragraph of *Results and Discussions* on page 6).

12. The sentence “At higher binding energies, the ellipses continue to shrink until they completely disappear for binding energies larger than **600** meV.” was replaced with “ At higher binding energies, the ellipses continue to shrink until they completely disappear for binding energies larger than **620** meV.” (Second paragraph on page 6)

13. We gave the value of m_x and m_y in unit of m_e according to the referee’s suggestion, and “ $m_x = 0.17 m_e$, $m_y = 0.85 m_e$ ” was added in the sixth paragraph of *Results and Discussions* on page 7.

14. We changed “3D map” to “3D $E(k_x, k_y)$ map” in the sentence “We also show a 3D map of the Rashba-like spin split bands as determined by ARPES in Fig. 4(a) ...” (in the seventh paragraph of *Results and Discussions* on page 8).
15. In the last paragraph of *Results and Discussions*, we changed the sentence “The obtained band splitting and dispersion match well with the calculated band dispersion along M–L direction as shown in Fig. 5(f).” to “The obtained band splitting along k_z direction can be qualitatively captured by the calculated results as shown in Fig. 5(f). The specific shape of the dispersions is a little different, which may be attributed to the poor momentum resolution along out-of-plane direction in ARPES measurements.”(page 8).
16. In the last paragraph of Results and Discussions, “So far, almost all of the reported Rashba splitting reside on 2D surface states. Although the Rashba band of BiTeI has bulk nature, it does not show k_z dependence, indicating that it arises from the near surface band-bending layer” was replaced with “So far, almost all of the reported Rashba bands only show spin splitting along in-plane momentum directions. Although the Rashba bands of BiTeI and GeTe show dispersion along out-of-plane direction (k_z), they remain spin degenerate along k_z for $k_x=k_y=0$ as sketched in Fig. 5(g), indicating a 2DRashba spin splitting.” (page 9).
17. The last sentence in Conclusion “In addition, the three-fold multiplicity of the M point directly implies a multi-pocket spin-split non-degenerate Fermiology” was replaced with “These results hence enrich our understanding of Rashba physics and inspire future exploration of new materials systems, which may host 3D Rashba spin texture and hold potential applications in spintronics.” (page 9).
18. We corrected some typos: “crossing pint” to “crossing point” on page 6; “Dresselhouse” to “Dresselhaus” on page 4 in the supplementary material.
19. We added “result from single-crystal XRD analysis” to the captions of Table 2 and Table 3 in the supplementary material.
20. We updated Table 2 in the supplementary material. The definitions of WP and U_{eq} were added.
21. “Fermi surface” was replaced with “constant energy contours” and “good agreement” was replaced with “reasonable agreement” in the last paragraph on page 4 in the supplementary material.
22. We added the estimation of inner potential V_0 in the supplementary material.

Reviewers' comments:

Reviewer #1 (Remarks to the Author):

The authors have improved the paper based on the referee comments. I think the paper is now getting to a place to be considered for publication. My main concern is again in relation to the theoretical calculations. Let me be more explicit in that regards

Based on the paper description and the reply that the authors, the theoretical description needs to be improved.

1) Using LDA and PBE, nowadays it is not enough to describe the properties of a material. It could be good for trends in databases but not for the characterization of a single material. Therefore, it would be important to see the band structure calculated by a more accurate methodology, such as many body or hybrid functionals. A mean field like functional could be not enough to capture the details.

2) The authors claim that to asses the quality of the DFT calculation is enough to calculate the cell parameter. This is not really true. Of course, if the purpose of the calculation is to obtain the cell parameter, that is the right methodology but if the desired property is the band structure, then, it has to be the cell parameters but also the exchange-correlation functional. The authors have used only two functionals, that we know, will not provide an accurate description to the details of the band structure, in particular over the Fermi energy. For these details, a hybrid or a meta-GGA functional is better.

3) How does the predicted Rashba parameter converge as a function of the cutoff and K-mess? how do the authors know that this quantity is fully converged? Experience in DFT has shown that any quantity of interest needs to be converged as a function of the energy cutoff (in PW like codes) and K-mesh.

4) The authors also say in the reply: "it is well known that LDA underestimates volumes of crystal while GGA overestimates those. Therefore, to assess the accuracy of DFT calculations, we should compare the theoretical lattice parameters and experimental ones".. I do not see why it has to be this way.... In reality, the argument provided by the authors, such as the agreement with the ARPES measurements is more physically sound....

5) Next time the authors provide a comparison between theory and experiment, they need to provide the details of the calculation, is it spin-polarized? does include spin-orbit? are the orbital contributions described in the figure? etc.

6) In the response to referee 2, the authors claim that: Since the Rashba energy is underestimated in the DFT calculations, the value of α_R is also underestimated in theoretical calculations, what do they conclude this? the prediction of all these quantities depends on the functional, the energy cutoff, etc.. but the authors claim is intrinsic to DFT, which is not correct.

7) Following as well to Referee 2, if now many of the claims have changed from good agreement to reasonable agreement, the authors need to stress the novelty of the results. In particular, the value of the theory, while the experiment is very clear.

After my concerns are clearly addressed I will support the paper publication in Nature Communications.

Reviewer #2 (Remarks to the Author):

The authors did an excellent job to improve their data, respectively their representation quality and answered all questions of mine and the other referees quite well.

I have only a few mandatory points and a few optional points to be considered prior to publication in Nature Communications.

Mandatory:

C10, C11, C12 and C18 have been answered in the rebuttal, but the answers did not make it into

the manuscript. This has to be changed, e.g.:

a) C10: Figure caption Fig 3: Add: "It has been checked that the experimental data in (b) exhibit the same spin chirality as the calculated data in d."

b) C11: page 7: "... exhibit a helical spin texture. The spin polarization in experiment is by a factor of 2-3 lower than in the calculations. This is typically observed [] and might be related to the k-space resolution in experiment, influence of the inelastic background, and sample inhomogeneity."

c) C12: Page S1: "... trigonal symmetry. Structure refinement has been done by evaluating the peak intensity ratios leading to the conclusion that the dominating structure is P31m. We summarize the information ..."

d) C18: Figure caption 2: "... labelled. Scale is linear as in all other images".

Optional:

a) The outlined perspectives emerging from the result of the authors are still rather blurry. I would skip the useless phrases at the end of the general intro "From an application point of view ... degree of freedom." and at the end of the total intro "Our observation ... spintronics." "...". They just transfer "I have no idea what it is good for, but I shout as loud as possible." Once mentioning applications at the very beginning of the manuscript and at the very end of the manuscript is enough, if there is no concrete idea on applications resulting from the achievements of the manuscript. The manuscript describes very good fundamental research without concrete application perspective. For the sake of honesty, one might go along with this.

b) Page 4 top: One could mention the VLEED detector. "... spin splitting in a binary compound: PtBi₂. We employed spin polarized ARPES with the recently developed VLEED-type detector enabling fast mapping of a complete spin-polarized E(k) map []. The obtained magnitude ... "

Reviewer #3 (Remarks to the Author):

I have read the revised manuscript "Giant anisotropic Rashba like spin splitting in PtBi₂" by Ya Feng and collaborators and author's response to Referee's criticism in the first review round. I maintain my view that the 3D anisotropic spin orbit splitting at M points of the Brillouin zone in the P31m phase of single crystal PtBi₂, resulting from cooperative Rashba and Dresselhaus interactions is an interesting new result worth of publication.

Regarding my comment on the poorer quality of the data showing the k_z dispersion of Figure 5 showing the energy momentum curves at different photon energies, Authors have made an effort to improve its readability. I understand that band dispersion along the k_z direction has poorer momentum resolution than in-plane curves. The use of second derivative plots, now in Figure S6 is very helpful to improve clarity.

My recommendation is to publish this paper in Nature Comms. as it will contribute to understand the unusual (and exotic) behaviors found in PtBi₂, and as such, will attract the interest of scientists working in Condensed Matter Physics, Material Sciences and Chemistry.

Response to Reviewer1

The authors have improved the paper based on the referee comments. I think the paper is now getting to a place to be considered for publication. My main concern is again in relation to the theoretical calculations. Let me be more explicit in that regards

Reply: We appreciate the referee for his/her positive evaluation of our work.

Based on the paper description and the reply that the authors, the theoretical description needs to be improved.

1) Using LDA and PBE, nowadays it is not enough to describe the properties of a material. It could be good for trends in databases but not for the characterization of a single material. Therefore, it would be important to see the band structure calculated by a more accurate methodology, such as many body or hybrid functionals. A mean field like functional could be not enough to capture the details.

Reply: We thank the Referee for this insightful suggestion. We performed the band calculations by using hybrid functionals (by using VASP) according to the Referee's suggestion and present the results in Fig. R1. We also include the calculated result by PBE and the experimental result for comparison. We can see that both the hybrid functionals and PBE methods can predict the Rashba splitting observed by the ARPES measurements. On the other hand, both methods cannot match the experimental results quantitatively.

Fig.R1 Band structure calculated by PBE(red lines) and HSE(blue lines) functionals along Γ -M- Γ . The experimental result is also included for comparison.

2) The authors claim that to asses the quality of the DFT calculation is enough to calculate the cell parameter. This is not really true. Of course, if the purpose of the calculation is to obtain the cell parameter, that is the right methodology but if the desired property is the band structure, then,

it has to be the cell parameters but also the exchange-correlation functional. The authors have used only two functionals, that we know, will not provide an accurate description to the details of the band structure, in particular over the Fermi energy. For these details, a hybrid or a meta-GGA functional is better.

Reply: We have followed the Reviewer's suggestions in the first-round report to do the lattice relaxation and found that the calculation with experimental lattice constants shows better agreement with ARPES measurements. As for the exchange-correlation functionals, the PBE calculation can give reasonable results comparing to experimental observations. We also performed HSE calculations which show similar band dispersions.

3) How does the predicted Rashba parameter converge as a function of the cutoff and K-mesh? how do the authors know that this quantity is fully converged? Experience in DFT has shown that any quantity of interest needs to be converged as a function of the energy cutoff (in PW like codes) and K-mesh.

Reply: We have carefully done the convergence test (k-mesh and Ecut) before we performed serious calculations. In the following, we list the data from convergence test, the k-mesh is 8x8x8 for Table R1 and the E-cutoff is 50 Ry for Table R2 and all parameters are for Γ -M line. One can find that the adopted k-mesh and E-cutoff in our paper is good enough to obtain a converged Rashba energy and Rashba parameter along Γ -M direction. The obtained Rashba parameter is $3.15 \text{ eV} \cdot \text{\AA}$, which is smaller than $4.36 \text{ eV} \cdot \text{\AA}$ in experiments.

In Page 7, after the first sentence of the third paragraph, we added a sentence: "The DFT calculated Rashba parameter along Γ M α_R is $3.15 \text{ eV} \cdot \text{\AA}$, smaller than the experimental value."

Table R1 Rashba energy, momentum offset and Rashba parameter as a function of E-cutoff with k-mesh 8x8x8.

Ecut(Ry)	ER(meV)	k0(1/\AA)	alphaR(eV · \AA)
45	69.0	0.0441	3.13
50	69.6	0.0442	3.15
55	69.6	0.0444	3.14
60	69.6	0.0442	3.15

Table R2 Rashba energy, momentum offset and Rashba parameter as a function of k-mesh with E-cutoff 50 Ry.

kmesh	ER(meV)	k0(1/\AA)	alphaR(eV · \AA)
6x6x6	69.5	0.0442	3.15
7x7x7	69.4	0.0442	3.14
8x8x8	69.6	0.0442	3.15
9x9x9	69.5	0.0442	3.15

The parameters in the kp model in the paper are obtained by fitting the DFT band structure with kp model. Due to the fitting errors, the obtained Rashba parameter $\alpha_1 - \beta_1 = 2.03$ is smaller than the value from DFT. Despite this, our model captures the main features of electronic structures

around M point.

4) The authors also say in the reply: "it is well known that LDA underestimates volumes of crystal while GGA overestimates those. Therefore, to assess the accuracy of DFT calculations, we should compare the theoretical lattice parameters and experimental ones".. I do not see why it has to be this way.... In reality, the argument provided by the authors, such as the agreement with the ARPES measurements is more physically sound....

Reply: We thank the referee for the comments. Yes, the band structure with experimental lattice parameters shows better agreement with ARPES data.

5) Next time the authors provide a comparison between theory and experiment, they need to provide the details of the calculation, is it spin-polarized? does include spin-orbit? are the orbital contributions described in the figure? etc.

Reply: We thank the referee for the suggestions. In our calculations, we performed calculations with spin-orbit coupling for nonmagnetic PtBi₂.

We have added this information in the third paragraph of Method:“, which achieved reasonable convergence of electronic structures. The calculations were done for nonmagnetic PtBi₂ with spin-orbit coupling .”

6) In the response to referee 2, the authors claim that: Since the Rashba energy is underestimated in the DFT calculations, the value of αR is also underestimated in theoretical calculations, what do they conclude this? the prediction of all these quantities depends on the functional, the energy cutoff, etc.. but the authors claim is intrinsic to DFT, which is not correct.

Reply: We thank the referee's suggestion for us to state this point more accurately. In our case, the calculated results do show smaller Rashba energy comparing to the experimental results.

7) Following as well to Referee 2, if now many of the claims have changed from good agreement to reasonable agreement, the authors need to stress the novelty of the results. In particular, the value of the theory, while the experiment is very clear.

After my concerns are clearly addressed, I will support the paper publication in Nature Communications.

Reply: The main novelty of our manuscript is the experimental discovery of 3D Rashba-like spin splitting at the M point. Although the theoretical calculation can't quantitatively match the experimental results very well, it correctly predicts the Rashba splitting around M point. Theoretical calculations also reveal the physical origination of the observed spin splitting in PtBi₂ and support the conclusion of this manuscript.

Response to Referee 2

The authors did an excellent job to improve their data, respectively their representation quality and answered all questions of mine and the other referees quite well.

I have only a few mandatory points and a few optional points to be considered prior to publication in Nature Communications.

Reply: We thank the referee for the positive comments of our revised manuscript and the good suggestions for us to further improve our manuscript. Please find our detailed reply to the comments below.

Reply to the referee's comments marked as “mandatory”:

C10, C11, C12 and C18 have been answered in the rebuttal, but the answers did not make it into the manuscript. This has to be changed, e.g.:

C1: *a) C10: Figure caption Fig 3: Add: “It has been checked that the experimental data in (b) exhibit the same spin chirality as the calculated data in d.”*

Reply: We thank the referee for the good suggestions. “It has been checked that the experimental data in **b** exhibits the same spin chirality as the calculated data in **d**.” was added to the caption of Fig. 3.

C2:b) C11: *page 7: “... exhibit a helical spin texture. The spin polarization in experiment is by a factor of 2-3 lower than in the calculations. This is typically observed [] and might be related to the k-space resolution in experiment, influence of the inelastic background, and sample inhomogeneity.”*

Reply: We have added “The spin polarization in experiments is by a factor of 2-3 lower than in the calculations. This was usually observed and might be related to the k-space resolution in experiments, influence of the inelastic background, and sample inhomogeneity.” behind the sentence “... exhibit a helical spin texture.” on page 7.

C3:c) C12: *Page S1: “... trigonal symmetry. Structure refinement has been done by evaluating the peak intensity ratios leading to the conclusion that the dominating structure is P31m. We summarize the information ...”*

Reply: Thanks for the suggestion. The sentences “Moreover, single crystal XRD analysis was performed on the sample and structure refinement has been done by evaluating the peak intensity ratios leading to the conclusion that the sample belongs to P31m space group. We summarize the information about its structure analysis in Table S1, Table S2, and Table S3.” has been added behind “The LEED pattern also supports that it belongs to trigonal symmetry.”

on page 1 of the Supplementary. And the sentence “We also summarized the information about its structure analysis in the following tables” was deleted.

C4:d) C18: Figure caption 2: “..labelled. Scale is linear as in all other images”.

Reply:“The color scale is linear as in all other images.” has been added at the end of the caption of Fig. 1d.

Reply to the referee's comments marked as “optional”:

C5:a) The outlined perspectives emerging from the result of the authors are still rather blurry. I would skip the useless phrases at the end of the general intro “From an application point of view degree of freedom.” and at the end of the total intro “Our observation ... spintronics.” “”. They just transfer “I have no idea what it is good for, but I shout as loud as possible.” Once mentioning applications at the very beginning of the manuscript and at the very end of the manuscript is enough, if there is no concrete idea on applications resulting from the achievements of the manuscript. The manuscript describes very good fundamental research without concrete application perspective. For the sake of honesty, one might go along with this.

Reply: We thank the referee for the comment and the suggestion. We deleted both the sentences “From an application point of view, a 3D Rashba material could unlock a whole class of unprecedented material functionalities, where the 3D spin current direction emerges as a tunable degree of freedom.” and “Our observation is therefore not only of fundamental interest for Rashba physics, but also of some potential applications in spintronics.” in the introduction.

C6:b) Page 4 top: One could mention the VLEED detector. “... spin splitting in a binary compound: PtBi₂. We employed spin polarized ARPES with the recently developed VLEED-type detector enabling fast mapping of a complete spin-polarized E(k) map []. The obtained magnitude ... ”

Reply: We thank the referee for the suggestion. “The spin-resolved ARPES employed the recently developed multichannel VLEED-type detector enabling fast mapping of a complete spin-polarized E(k) map [51]” has been added behind the sentence “In this work, we report the discovery... in a binary compound: PtBi₂” in the last paragraph of the Introduction.

Response to Referee 3

Reviewer #3 (Remarks to the Author):

I have read the revised manuscript “Giant anisotropic Rashba like spin splitting in PtBi₂” by Ya Feng and collaborators and author's response to Referee's criticism in the first review round. I maintain my view that the 3D anisotropic spin orbit splitting at M points of the Brillouin zone in the P31m phase of single crystal PtBi₂, resulting from cooperative Rashba and Dresselhaus interactions is an interesting new result worth of publication.

Regarding my comment on the poorer quality of the data showing the k_z dispersion of Figure 5 showing the energy momentum curves at different photon energies, Authors have made an effort to improve its readability. I understand that band dispersion along the k_z direction has poorer momentum resolution than in-plane curves. The use of second derivative plots, now in Figure S6 is very helpful to improve clarity.

My recommendation is to publish this paper in Nature Comms. as it will contribute to understand the unusual (and exotic) behaviors found in PtBi₂, and as such, will attract the interest of scientists working in Condensed Matter Physics, Material Sciences and Chemistry.

Reply: We thank the referee for his/her recommendation for publication of our manuscript in Nature Communications.

List of changes:

1. In the third paragraph of Method: “, which achieved reasonable convergence of electronic structures. The calculations were done for nonmagnetic PtBi₂ with spin-orbit coupling .” was added behind the sentence “The Brillouin zone was sampled in k space within the Monkhorst-Pack scheme[55] with a kmesh of 8x8x8.”
2. Page 7, after the first sentence of the third paragraph, we added “The DFT calculated Rashba parameter along $\Gamma M \alpha_R$ is 3.15 eV · Å, smaller than the experimental value.”
3. “It has been checked that the experimental data in **b** exhibits the same spin chirality as the calculated data in **d**.” was added to the caption of Fig. 3.
4. “The spin polarization in experiments is by a factor of 2-3 lower than in the calculations. This was usually observed and might be related to the k-space resolution in experiments, influence of the inelastic background, and sample inhomogeneity.” was added behind “... exhibit a helical spin texture.” on page 7.
5. In the first paragraph of the supplementary material:
“Moreover, single crystal XRD analysis was performed on the sample and structure refinement has been done by evaluating the peak intensity ratios leading to the conclusion that the sample belongs to P31m space group. We summarize the information about its structure analysis in Table S1, Table S2, and Table S3.” was added behind “The LEED pattern also supports that it belongs to trigonal symmetry.”
The sentence “We also summarized the information about its structure analysis in the following tables” was deleted.
6. “The color scale is linear as in all other images.” was added at the end of the caption of Fig. 1d.
7. In the third paragraph of the introduction.
“From an application point of view, a 3D Rashba material could unlock a whole class of unprecedented material functionalities, where the 3D spin current direction emerges as a tunable degree of freedom.” was deleted.
8. In the last paragraph of the introduction:
“The spin-resolved ARPES employed the recently developed multichannel VLEED-type detector enabling fast mapping of a complete spin-polarized E(k) map [51]” was added behind the sentence “In this work, we report the discovery... in a binary compound: PtBi₂”
And the sentence “Our observation is therefore not only of fundamental interest for Rashba physics, but also of some potential applications in spintronics.” was deleted.

REVIEWERS' COMMENTS:

Reviewer #1 (Remarks to the Author):

The authors have successfully responded to each one of my concerns. Now the support obtained from new calculations and the clarification of different issues has now strengthened the paper and the conclusions. I really support the publication of this paper in Nature Communications.

Reviewer #2 (Remarks to the Author):

everything has been answered.
Manuscript is ready for publication